

Atmospheric
Measurement
Techniques

# Measurement of $NO_x$ and $NO_y$ with a thermal dissociation cavity ring-down spectrometer (TD-CRDS): instrument characterisation and first deployment

Nils Friedrich[1], Ivan Tadic[1], Jan Schuladen[1], James Brooks[2], Eoghan Darbyshire[2], Frank Drewnick[3], Horst Fischer[1], Jos Lelieveld[1], and John N. Crowley[1]

[1]Atmospheric Chemistry Department, Max Planck Institute for Chemistry, Mainz, 55128, Germany
[2]Centre for Atmospheric Science, University of Manchester, Manchester, M13 9PL, UK
[3]Particle Chemistry Department, Max Planck Institute for Chemistry, Mainz, 55128, Germany

**Correspondence:** John N. Crowley (john.crowley@mpic.de)

**Abstract.** We present a newly constructed, two-channel thermal dissociation cavity ring-down spectrometer (TD-CRDS) for the measurement of $NO_x$ ($NO + NO_2$), $NO_y$ ($NO_x + HNO_3 + RO_2NO_2 + N_2O_5$ TS1 etc.), $NO_z$ ($NO_y - NO_x$) and particulate nitrate (pNit). $NO_y$-containing trace gases are detected as $NO_2$ by the CRDS at 405 nm following sampling through inlets at ambient temperature ($NO_x$) or at 850 °C ($NO_y$). In both cases, $O_3$ was added to the air sample directly upstream of the cavities to convert NO (either ambient or formed in the 850 °C oven) to $NO_2$. An activated carbon denuder was used to remove gas-phase components of $NO_y$ when sampling pNit. Detection limits, defined as the $2\sigma$ precision for 1 min averaging, are 40 pptv for both $NO_x$ and $NO_y$. The total measurement uncertainties (at 50 % relative humidity, RH) in the $NO_x$ and $NO_y$ channels are 11 %+10 pptv and 16 %+14 pptv for $NO_z$ respectively. Thermograms of various trace gases of the $NO_z$ family confirm stoichiometric conversion to $NO_2$ (and/or NO) at the oven temperature and rule out significant interferences from $NH_3$ detection ($< 2$ %) or radical recombination reactions under ambient conditions. While fulfilling the requirement of high particle transmission ($> 80$ % between 30 and 400 nm) and essentially complete removal of reactive nitrogen under dry conditions ($> 99$ %), the denuder suffered from $NO_x$ breakthrough and memory effects (i.e. release of stored $NO_y$) under humid conditions, which may potentially bias measurements of particle nitrate.

Summertime $NO_x$ measurements obtained from a ship sailing through the Red Sea, Indian Ocean and Arabian Gulf ($NO_x$ levels from $< 20$ to 25 ppbv) were in excellent agreement with those taken by a chemiluminescence detector of NO and $NO_2$. A data set obtained locally under vastly different conditions (urban location in winter) revealed large diel variations in the $NO_z$ to $NO_y$ ratio which could be attributed to the impact of local emissions by road traffic.

## 1 Introduction

### 1.1 Atmospheric $NO_x$ and $NO_y$

Total reactive nitrogen $NO_y$ ($= NO_x + NO_z$) consists of nitrogen oxide, NO; nitrogen dioxide, $NO_2$ ($NO + NO_2 = NO_x$); and their reservoir species, $NO_z$ ($NO_3 + 2N_2O_5 + HNO_3 + HONO + RONO_2 + RO_2NO_2 + XONO_2 + XNO_2 + pNit$), where X is a halogen atom. HCN and $NH_3$ are generally not considered to be components of $NO_y$ (Logan, 1983).

The formation of both peroxy nitrates (PNs; $RO_2NO_2$) and alkyl nitrates (ANs; $RONO_2$) requires the presence of organic peroxy radicals ($RO_2$), which are formed by processes such as the reaction of OH radicals with volatile organic compounds (VOCs) and oxygen (Reaction R1). $RO_2$ radicals subsequently react with $NO_2$ or NO to form peroxy nitrates (PNs; $RO_2NO_2$) or alkyl nitrates (ANs; $RONO_2$, Re-

*Please note the remarks at the end of the manuscript.*

actions R2 and R3). Reaction (R3) competes with the formation of an alkoxy radical (RO) and the oxidation of NO to $NO_2$ (Reaction R4), which consumes the dominant fraction of $RO_2$. The branching ratio between these two pathways depends on atmospheric conditions such as pressure and temperature and on the structure and length of the organic backbone (Lightfoot et al., 1992). $HNO_3$ is produced mainly via the reaction of $NO_2$ with OH (Reaction R5).

$$OH + RH + O_2 \rightarrow RO_2 + H_2O \qquad (R1)$$
$$RO_2 + NO_2 + M \rightarrow RO_2NO_2 + M \qquad (R2)$$
$$RO_2 + NO + M \rightarrow RONO_2 + M \qquad (R3)$$
$$RO_2 + NO \rightarrow RO + NO_2 \qquad (R4)$$
$$OH + NO_2 + M \rightarrow HNO_3 + M \qquad (R5)$$

The lifetimes of peroxy nitrates in the low troposphere are mainly governed by the temperature. PNs with an additional acyl group (PANs), such as peroxyacetyl nitrate (PAN), are generally more stable than PNs without an acyl group (e.g. pernitric acid, $HO_2NO_2$), which are observed only in cold regions (Slusher et al., 2002). Thus, of the peroxy nitrates, only PANs are considered able to act as transportable reservoirs for $NO_x$. At higher altitudes in the troposphere (above ca. 7 km) photolysis becomes the most important loss process for PAN, while the reaction with OH is negligible throughout the troposphere (Talukdar et al., 1995).

The absence of photolysis reactions and low levels of the OH radical at night-time provide alternative pathways for the formation of $NO_z$ species. $NO_2$ is oxidised by $O_3$ to produce the nitrate radical $NO_3$, which exists in thermal equilibrium with $N_2O_5$ (Reactions R6 and R7). The reaction of $NO_3$ with hydrocarbons represents a night-time source of alkyl nitrates (Reaction R8), and $N_2O_5$ can be hydrolysed on aqueous aerosol, resulting in the formation of $HNO_3$ (Reaction R9) and $ClNO_2$ (Reaction R10) if particulate chloride is available (Finlayson-Pitts et al., 1989). TS2

$$NO_2 + O_3 \rightarrow NO_3 + O_2 \qquad (R6)$$
$$NO_3 + NO_2 \rightleftharpoons N_2O_5 \qquad (R7)$$
$$NO_3 + R=R(+O_2) \rightarrow RONO_2 \qquad (R8)$$
$$N_2O_5 + H_2O \rightarrow 2HNO_3 \qquad (R9)$$
$$N_2O_5 + Cl^- \rightarrow ClNO_2 + NO_3^- \qquad (R10)$$

Nitric acid formation via the reaction of $NO_2$ and OH (Reaction R5), followed by wet or dry deposition of $HNO_3$, is considered to be the dominant daytime loss process for atmospheric $NO_x$ (Roberts, 1990), although the reduction of $NO_x$ may result in an increasingly important role for organic nitrates (e.g. in the USA; Romer Present et al., 2020). As some organic nitrates have longer lifetimes than $HNO_3$, the atmospheric transport of $NO_x$ to remote locations would lead to a more even distribution of $NO_x$, instead of hotspots in polluted regions close to emission sources. Atmospheric removal processes for ANs include oxidation by OH or $O_3$

(which may lead to a loss of the nitrate functionality), deposition to the Earth's surface and photolysis. Additionally, partitioning into the aerosol phase is possible for large and multifunctional ANs (Perring et al., 2013). Alkyl nitrates possessing no further functionality (e.g. double bonds or hydroxyl groups) can be unreactive and have long lifetimes (Talukdar et al., 1997). On the global average, $RONO_2$ has a lifetime of close to 3 h (2.6–3 h) with $\sim 30\%$ being lost by hydrolysis (Zare et al., 2018).

The formation of $NO_z$ in the lower atmosphere reduces the $NO_x$ lifetime, and the partitioning of $NO_y$ into $NO_x$ and $NO_z$ can provide information about the chemical history of an air mass (Day et al., 2002; Wild et al., 2014). In regions impacted by biogenic emissions, the sources and sinks of ANs account for a large fraction of $NO_x$ lost both during the day and night and, thus, control the lifetime of $NO_x$ (Romer et al., 2016; Sobanski et al., 2017).

Laboratory experiments have shown that particulate nitrates (pNits) are formed at high yields in the atmospheric degradation of terpenoids in the presence of $NO_x$ and play an important role in the formation and growth of secondary organic aerosol (SOA; Ng et al., 2017; Ammann et al., 2019). This has been confirmed in field studies, which provide evidence for the partitioning of organic nitrate to the aerosol phase both during day- and night-time (Rollins et al., 2012; Fry et al., 2013; Palm et al., 2017) with the formation of highly functionalised molecules and large contributions (up to 25 %) of particulate organic nitrates to the total aerosol mass (Xu et al., 2015; Lee et al., 2016; Huang et al., 2019).

## 1.2 Detection of $NO_x$

Methods for the detection of NO and $NO_2$ include chemiluminescence (CLD), differential optical absorption spectroscopy (DOAS), laser-induced fluorescence (LIF) and cavity ring-down spectroscopy (CRDS). A description and intercomparison of these methods is given in Fuchs et al. (2010), and we restrict the following discussion to an outline of the basic principles. The CLD method detects NO by chemiluminescent emission in its reaction with $O_3$; detection of ambient $NO_2$ by CLD follows its catalytic or photolytic conversion to NO. The best CLD devices have detection limits for NO and $NO_2$ in the single-digit parts per trillion by volume (pptv) range (Hosaynali Beygi et al., 2011; Reed et al., 2016; Tadic et al., 2020). Detection of $NO_2$ via LIF involves photoexcitation in its visible absorption band at wavelengths $> 400$ nm and detection of fluorescent emission at wavelengths $> 600$ nm, with detection limits of the order of parts per trillion by volume achieved for an integration time of a few seconds (Day et al., 2002; Javed et al., 2019). The structured spectrum of $NO_2$ between $\approx 400$ and 600 nm is used to detect light absorption by ambient $NO_2$ by DOAS, using either broadband light sources (long-path DOAS, with a path length of more than a few kilometres) or natural sun-

light (Platt et al., 1979; Leser et al., 2003; Pohler et al., 2010; Merten et al., 2011).

The CRDS detection method for $NO_2$ also utilises its visible absorption spectrum, with high sensitivity being reached by achieving very long path lengths for optical extinction in an optical resonator (see Sect. 2.1). Limits of detection for $NO_2$ with CRDS of $< 20$ pptv with a 1 s integration time have been reported (Wild et al., 2014). NO can be detected as $NO_2$ following its oxidation by $O_3$ (Reaction R6; Fuchs et al., 2009).

## 1.3 Detection of $NO_y$

The first $NO_y$ measurements were based on the conversion of all reactive nitrogen trace gases (apart from NO) to NO on catalytic metal surfaces of gold at $\sim 300$–$320\,^\circ$C or of molybdenum oxide (MoO) at $\sim 350$–$400\,^\circ$C (Fahey et al., 1985; Williams et al., 1998), with subsequent CLD detection of NO. Au converters were designed to exclude particulate nitrates, whereas MoO set-ups aimed at a response towards pNit (Williams et al., 1998). In recent years, the thermal decomposition of $NO_z$ to $NO_2$ has been employed to detect $NO_z$ using inlets held at temperatures high enough ($> 650$–$700\,^\circ$C) to thermally dissociate the most strongly bound reactive nitrogen trace gas, $HNO_3$, to $NO_2$ (Day et al., 2002; Rosen et al., 2004; Wooldridge et al., 2010; Perring et al., 2013; Wild et al., 2014) and/or using multiple inlets at intermediate temperatures (Paul et al., 2009; Paul and Osthoff, 2010; Sadanaga et al., 2016; Sobanski et al., 2016; Thieser et al., 2016). Following thermal decomposition, the $NO_2$ product can be detected using LIF (Day et al., 2002, 2003; Rosen et al., 2004; Murphy et al., 2006; Wooldridge et al., 2010) or cavity-enhanced absorption spectroscopy (Paul et al., 2009; Wild et al., 2014; Sadanaga et al., 2016; Sobanski et al., 2016; Thieser et al., 2016). These techniques are impacted to various degrees by secondary reactions at high temperatures, including the loss of $NO_2$ via recombination with $\alpha$-carbonyl peroxy radicals or reaction with O TS3 atoms (formed by the thermolysis of ambient $O_3$) and the generation of extra $NO_2$ from the oxidation of NO via reactions with peroxy radicals (Day et al., 2002; Sobanski et al., 2016; Thieser et al., 2016; Womack et al., 2017). Measures to reduce potential measurement artefacts and avoid excessive data correction include operation at low pressures (Day et al., 2002; Womack et al., 2017) and the addition of surfaces to scavenge peroxy radicals (Sobanski et al., 2016). Nonetheless, data correction may still be necessary and may involve laboratory characterisation and chemical simulation of the chemical reactions within the heated inlet (Sobanski et al., 2016; Thieser et al., 2016).

In this paper we present a two-channel TD-CRDS instrument for the detection of $NO_x$, $NO_y$, $NO_z$ and pNit that overcomes these limitations. Compared with the set-ups described by Thieser et al. (2016), the following changes were implemented: (1) the addition of $O_3$ for $NO_x$ detection,

(2) a higher oven temperature (to detect $HNO_3$) and located CE1 directly at the front of the inlet, and (3) the use of a charcoal denuder for separate measurement of pNit and gas-phase $NO_z$. The addition of $O_3$ (after the TD inlet) ensures that we detect NO as well as $NO_2$ and, thus, removes bias caused by factors such as the pyrolysis of $O_3$ and reactions of $O(^3P)$ which reduce $NO_2$ to NO.

## 2 Experimental

Our TD-CRDS instrument consists of two identically constructed cavities to monitor $NO_2$ at 405 nm which are largely unchanged compared to those described by Thieser et al. (2016). In the present set-up, the two cavities are connected to three different inlets. One cavity monitors $NO_x$ via an inlet at ambient temperature, and the second samples air via one of two heated inlets (one equipped with a denuder, see below), thereby monitoring either $NO_y$ or particle nitrate. A schematic diagram (not to scale) of the instrument is given in Fig. 1.

## 2.1 CRDS operation principals

The optical resonator consists of two mirrors (1 m radius of curvature) with a nominal 0.999965 reflectivity at 405 nm (Advanced Thin Films), which are mounted 70 cm apart. The cavity volumes are defined by Teflon (FEP) coated DURAN glass tubes with an inner diameter of 10 mm.

Under normal operating conditions each cavity samples $3.0\,L$ (STP) $min^{-1}$ (slpm; standard litres per minute) of ambient air (where STP refers to $0\,^\circ$C and 1013 hPa). Additional purge flows (0.14 slpm dry synthetic air) are introduced directly in front of each mirror to prevent surface degradation by atmospheric trace gases. The cavities are operated at pressures of 540 to 580 Torr (1 Torr $= 1.333$ hPa), resulting in a residence time of $\sim 1.2$ s. A 405 nm laser light (square-wave modulated at 1666 Hz and a 50 % duty cycle) is provided by a laser diode (LASER COMPONENTS), the emission of which is coupled into an optical fibre with a Y piece for splitting into both cavities. Temperature and current control of the laser diode are achieved by a Thorlabs ITC502 control unit. The laser emission spectrum is monitored continuously by coupling scattered light from one of the cavity mirrors into a 3648 pixel CCD (charge-coupled device) spectrograph (OMT, $\sim 0.1$ nm resolution).

The intensity of light exiting the cavity is measured with a photomultiplier (Hamamatsu Photonics), with ring-up and ring-down profiles recorded by a digital oscilloscope (Pico-Scope 3000). $NO_2$ mixing ratios are derived from the decay constant ($k$ or $k_0$) describing the exponential decrease in light intensity after the laser has been switched off:

$$[NO_2] = \frac{l}{d} \cdot \frac{1}{\sigma c} (k - k_0), \tag{1}$$

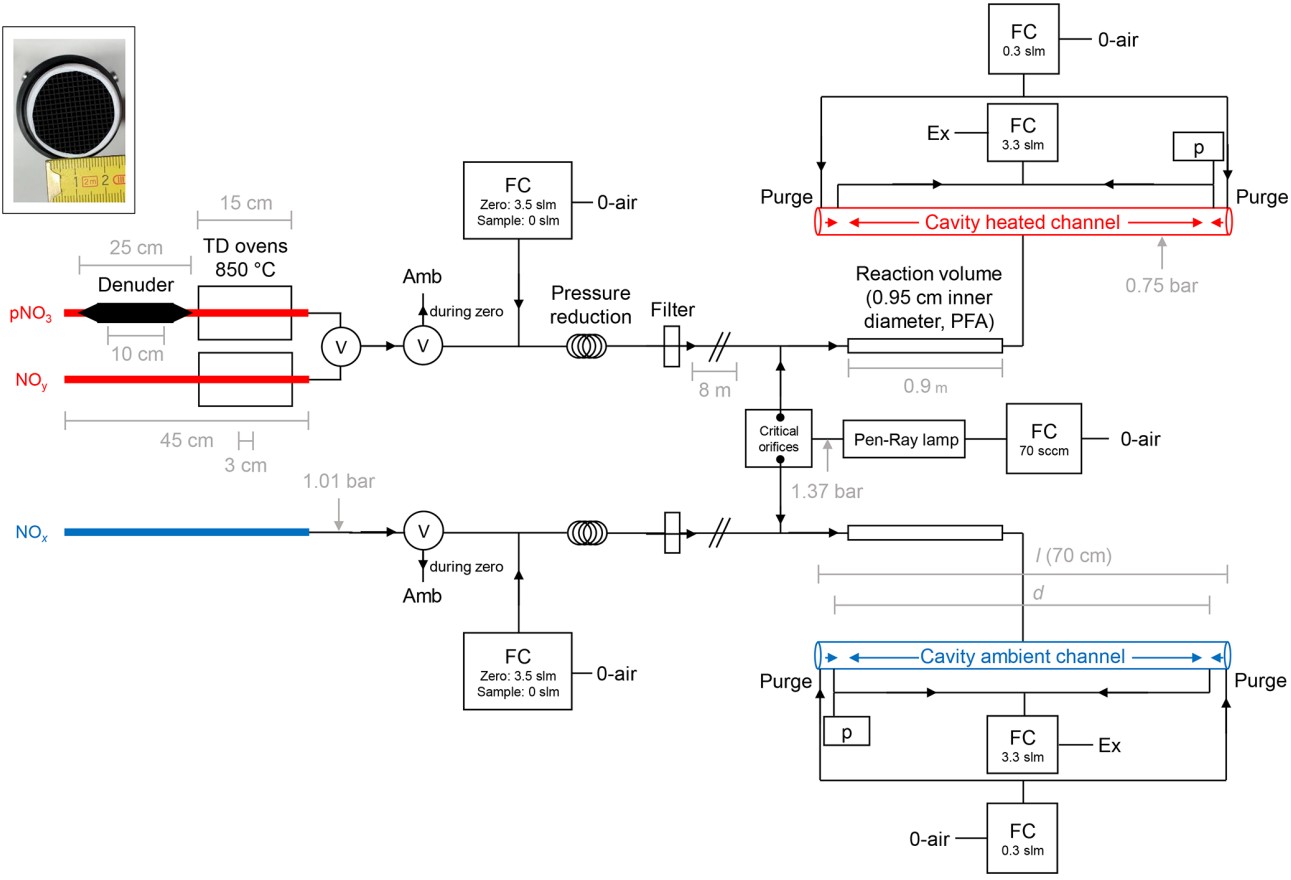

**Figure 1.** Schematic diagram of the TD-CRDS instrument (not to scale). NO$_y$ and pNit are detected via the heated channel, and NO$_x$ is detected via the ambient channel. Ozone is generated via a Pen-Ray lamp (185 nm) and serves to convert NO to NO$_2$. TD denotes thermal dissociation, and FC denotes flow controller. The flows listed are those used under normal operating conditions. p denotes a pressure sensor, Ex denotes a membrane pump and exhaust, Amb denotes ambient air and V denotes an electronically switchable PTFE valve. "Filter" is a PTFE filter, 2 µm pore size. O-air refers to zero air CE2. The inset (photo) shows the honeycomb structure of the activated carbon denuder. The critical orifices have diameters of $\approx 0.05$ mm. ("slm" is used to refer to standard litres per minute in this figure.)

where $c$ is the speed of light, $\sigma$ is the effective absorption cross section of NO$_2$ over the emission spectrum of the laser (Vandaele et al., 2002), and $k$ and $k_0$ are the decay constant with and without NO$_2$ present in the cavity respectively. Thus, $k_0$ is defined by the mirror reflectivity and light scattering by the dry, synthetic air.

The ratio $l/d$ accounts for the difference between the physical length of the cavity ($l$) and the effective optical path length ($d$) in which NO$_2$ is present as well as for dilution effects. $d$ is shorter than $l$ due to the purge flows of zero air in front of the mirrors, and a value of $l/d = 0.98 \pm 0.01$ TS4 was determined by adding a constant flow of NO$_2$ and varying the purge-gas flow rate (Schuster et al., 2009; Thieser et al., 2016). $k_0$ is typically determined every 5 min (for 1 min) by overflowing the inlets with zero air from a commercial zero air generator (CAP 180, Fuhr GmbH) attached to a source of compressed ambient air. PTFE filters (47 mm diameter, 2 µm pore size) prevent particles from entering the cavities. The filter's efficiency, tested with laboratory air contain-

ing $1.8 \times 10^3$ particles cm$^{-3}$ and a CPC (TSI 3025 A), was $> 98 \%$.

Raw data sets (i.e. ring-down constants) undergo a few basic corrections before further analysis:

1. $k_0$ is interpolated onto the $k$ time grid. The first three data points after switching from sampling to zeroing are discarded in order to enable the stabilisation of the zero signal. The remaining data points of each zero cycle are averaged. Finally, a linear interpolation between the averaged $k_0$ values is performed, allowing for the subtraction of $k_0$ for each individual data point.

2. Depending on conditions of flow, the pressure and the inlet set-up (see Sect. 2.2 and 2.3), changes in flow resistance between the zeroing and sampling periods result in slight changes in the cavity pressure. The resulting change in Rayleigh scattering of the 405 nm light owing to a pressure change of 6.5 Torr was found to be equivalent to a change of ca. 300 pptv in the NO$_2$ mixing ratio,

which is in accordance with earlier experiments using previous versions of this instrumental set-up (Thieser et al., 2016). We have also used an alternative set-up in which the inlet is overflowed with zero air added close to the tip of the inlet (downstream TS5 of the oven); this reduces the pressure difference but has the disadvantage that hot air is blown out of the instrument when zeroing, which may interfere with co-located inlets. The addition of zero air upstream of the quartz inlets would remove this problem, but it would also increase the complexity of the inlet and potentially result in the loss of sticky molecules such as $HNO_3$.

3. A further correction is associated with the difference in the Rayleigh scattering coefficient between dry air (during zeroing) and humid air (whilst taking ambient measurements). This effect was corrected using the $H_2O$ scattering cross sections reported by Thieser et al. (2016), leading, for example, to a correction of 116 pptv at 70 % RH and 20 °C.

## 2.2 Detection of $NO_x$

In order to measure NO (which does not absorb at 405 nm), it is converted to $NO_2$ by reaction with an excess of ozone ($O_3$); the $O_3$ was generated by passing zero air over a Hg Pen-Ray lamp emitting at 185 nm, which was housed inside a glass vessel at ca. 980 Torr pressure. The gas stream containing $O_3$ is split up equally by critical orifices and directed into two identical reaction volumes made of 88 cm long PFA tubing (1/2 inch outer diameter, residence time 1.05 s). The concentration of $O_3$ (monitored by a commercial monitor, Model 202, 2B Technologies) was optimised in laboratory experiments in which the efficiency of the conversion of NO to $NO_2$ was varied by changing the flow of air over the Pen-Ray lamp. The maximum concentration of $NO_2$ (corresponding to 96 % of the NO in the gas bottle) was observed when the flow over the Pen-Ray lamp was between 60 and 80 $cm^3$ (STP) $min^{-1}$ (hereafter sccm), which resulted in 19 ppmv $O_3$ in the reaction volumes. This result could be confirmed by numerical simulation (see Table S1 in the Supplement) of the reactions involved in the formation and loss of $NO_2$ when NO reacts with $O_3$. According to the simulation, the maximum conversion of NO to $NO_2$ during the 1.05 s residence time occurs between ca. 12 and 20 ppmv $O_3$. The conversion efficiency decreases at higher $O_3$ concentrations due to the formation of $N_2O_5$ and $NO_3$. The results from the experiments to determine the optimum parameters for $O_3$ generation are summarised in Fig. S1.

For $NO_2$, the performance of the instrument was first described by Thieser et al. (2016), who reported a measurement uncertainty of 6 % + (20 pptv × RH/100) that was dominated by uncertainty in the effective cross section of $NO_2$ and the wavelength stability of the laser diode. The $NO_x$ detection limit of 40 pptv ($2\sigma$, 1 min average) for the present instru-

ment (laboratory conditions) was derived from an Allan variance analysis and is worse than that reported by Thieser et al. (2016) (6 pptv at 40 s) due to degradation of the mirror reflectivity. Corrections applied to take humidity and pressure changes into account are discussed in Sect. 2.1. The total uncertainty in $NO_y$ will depend on the uncertainty in the conversion of both gaseous and particulate nitrate to $NO_x$ and, thus, depends on the individual components of $NO_y$ in the air sampled. For purely gaseous $NO_y$, the major problem is likely to be related to the loss of sticky molecules at the inlet, and we choose to quote a "worst case" uncertainty of 15 %.

## 2.3 Thermal dissociation inlets: detection of $NO_y$

The thermal dissociation inlets used to dissociate $NO_y$ to $NO_2$ are quartz tubes housed in commercial furnaces (Carbolite, MTF 10/15/130). The oven temperature was regulated with a custom-made electronic module, which enabled spatial separation between the heating elements and insulation and the control electronics. The distance between the heated section of the quartz tubes and the point at which air was taken into the inlet was kept short (ca. 30 cm) in order to minimise losses of trace gases with a high affinity for surfaces, especially $HNO_3$ (Neuman et al., 1999). Experiments characterising the thermal conversion of various trace gases to $NO_2$ are described in Sect. 3.1. An electronic, PTFE three-way valve (Neptune Research, Inc., type 648T032, orifice diameter 4 mm) under software control switches between the two heated inlets, one of which is equipped with a denuder. Memory effects on the valves' surfaces were not observed for $NO_2$. Bypassing the valve under normal sampling conditions led to a 0.6 Torr pressure change. The sampling flow through both heated inlets is 3.0 slpm. When sampling ambient air via the denuder, we expect to remove all gas-phase $NO_y$ components and, thus, measure only particulate nitrate (pNit). Experiments to characterise the transmission of the denuder for particles and various trace gases are presented in Sect. 3.3.

## 2.4 Active carbon denuder

The active carbon denuder (Dynamic AQS) has a honeycomb structure with 225 quadratic channels (1 mm × 1 mm) of 10 cm length in a cylindrical form (diameter 3 cm) which is housed inside an aluminium casing with 1/2 inch connections (see Fig. 1). The geometric surface area of the denuder is ∼ 45 $cm^2$. Assuming a specific surface area for activated carbon of 1000 $m^2 g^{-1}$ (Atsuko et al., 1996), we calculate a BET (Brunauer–Emmett–Teller, Brunauer et al., 1938) surface area of the order of $10^8$ $cm^2$.

## 2.5 Chemicals

A stock, liquid sample of PAN in $n$-tridecane (> 98 %, Alfa Aesar) was synthesised according to the procedures described by Gaffney et al. (1984) and Talukdar et al. (1995).

Samples of lower concentration (as used in the experiments described below) were produced by diluting the original sample with additional *n*-tridecane. Acetone (> 99 %), isopropyl nitrate (> 98.0 %) and (R)-(+)-limonene (97 %) were obtained from Sigma-Aldrich. An ammonia permeation source (324 ng min$^{-1}$) was supplied by VICI Metronics. Methanol (> 99.9 %) was acquired from Merck, isoprene (98 %, stabilised) from Acros Organics, and ethanol from Martin and Werner Mundo oHG. Both nitric acid (65 %) and $\beta$-pinene (pure) were obtained from Carl Roth. $N_2O_5$ crystals were synthesised according to Davidson et al. (1978) by reacting NO (5 %) with excess $O_3$ in a glass reactor. $O_3$ was produced via electrical discharge through $O_2$ using a commercial ozone generator (Ozomat Com, Anseros). The crystals were trapped and stored at $-78$ °C in dry ice and ethanol.

## 3  Results and discussion

### 3.1  Trace gas thermograms

The fractional conversion of $NO_z$ to $NO_2$ in the TD inlets was investigated in a series of experiments in which constant flows of various $NO_z$ trace gases were passed through the heated inlet (bypassing the denuder) while the temperature was varied and $NO_2$ was monitored. $NO_x$ impurity levels were determined either via the simultaneous operation of the $NO_x$ channel or via the $NO_y$ channel mixing ratio at room temperature, before and after heating the inlet. By inserting a thermocouple into the middle part of the heated section under normal sampling conditions, we were able to show that the temperature of the gas was $\approx 80$ °C lower than that indicated by the oven's internal temperature sensor in the 200–300 °C temperature range and about 40 °C lower at a set temperature of 600 °C (see Fig. S3a). We were unable to measure the temperatures of the gas stream at oven temperatures above about 600 °C, and we refer only to the temperature indicated by the internal sensor of the oven throughout the paper.

In the following, we show that the thermograms (plots of fractional dissociation of $NO_z$ to $NO_x$ versus temperature) which we measure with this instrument are broader and are shifted in temperature compared with other examples found in the literature, including those from this laboratory. As this instrument is built for the measurement of $NO_y$ and is not intended for separate measurement of species such as PNs, ANs and HNO$_3$, overlap of the individual thermograms does not represent a problem.

### 3.1.1  PAN

A stream of 200 sccm synthetic air was used to elute a constant supply of gaseous PAN from its solution (held at a constant temperature of 0 °C in a glass vessel) into the CRDS inlet. The thermogram is presented in Fig. 2, and the absolute $NO_2$ mixing ratios are depicted in Fig. S2a. In this experiment the maximum amount of $NO_2$ observed (at tem-

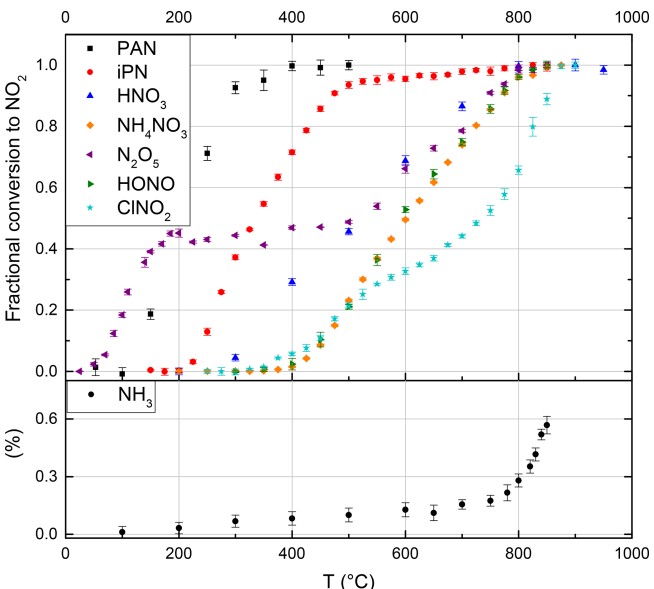

**Figure 2.** Thermograms of the $NO_z$ species PAN, iPN (isopropyl nitrate), HNO$_3$, NH$_4$NO$_3$, N$_2$O$_5$, HONO, ClNO$_2$ and the potential interference from NH$_3$ (without added O$_3$). The NH$_3$ fractional conversion is calculated relative to the calibrated output of the permeation source, whereas all other species are calculated relative to the observed mixing ratio at maximum conversion. Error bars are derived from the standard deviations during the averaging intervals. At the set temperature of 850 °C, PAN, iPN, NH$_4$NO$_3$, N$_2$O$_5$ (x2), HONO and HNO$_3$ are converted quantitatively to NO$_2$, while the NH$_3$ interference is negligible under typical ambient conditions.

peratures > 400 °C) was 2.2 ppbv. At temperatures < 100 °C, there was no measurable thermal decomposition of PAN to $NO_2$. Increasing the temperature from 100 °C to 300 °C resulted in a sharp increase in $NO_2$ which flattened off at temperatures > 380 °C. We conclude that PAN is stoichiometrically converted to $NO_2$ at temperatures above 400 °C in our oven. The steepest part of the isotherm at $\sim 200$ °C, i.e. 50 % conversion of PAN to $NO_2$, is therefore shifted by ca. 80 °C compared with those reported in the literature by Wild et al. (2014), Thieser et al. (2016) and Sobanski et al. (2016). This is a consistent feature of our TD ovens and is related to the short time available for thermal decomposition (see below) and a significantly lower gas temperature than indicated by the oven's internal temperature sensor.

### 3.1.2  Isopropyl nitrate

A 10 L stainless-steel canister containing 10.3 ppmv of isopropyl nitrate (iPN) at a pressure of 4 bar N$_2$ was prepared using a freshly vacuum-distilled liquid sample utilising standard manometric methods. $NO_x$ impurities were $\sim 4.7$ ppbv, although we note that diluted iPN stored in stainless-steel canisters for periods of several weeks degrades to form NO$_2$ and HNO$_3$.

The thermogram is displayed in Fig. 2, and the absolute concentrations are shown in Fig. S2b. Based on the mixing ratio of iPN in the canister and the dilution flows, 10.7 ppbv represents $(101 \pm 11)$ % conversion. The shaded area around the expected iPN mixing ratio in Fig. S2b signifies the uncertainty of this value based on the propagation of the errors during the manometric and dilution procedures (2 % for flow rates, 5 % for pressures measured with digital pressure gauges and 10 % for the last dilution step using the analogue pressure gauge of the canister).

Between 550 and 850 °C, we observe a weak increase in $NO_2$ from 10.7 to 11.2 ppbv, which is likely due to small amounts of $HNO_3$ in the sample. For iPN, the temperature at 50 % conversion is 50 °C higher than those reported by Thieser et al. (2016) and Sobanski et al. (2016). Wild et al. (2014) employed a gaseous mixture of different alkyl nitrates and also observed an initial increase in $NO_2$ (up to 80 % conversion) for temperatures $< 300$ °C, followed by a slower increase up to 800 °C. The alkyl nitrates thermogram of Wild et al. (2014) has been included into Fig. S2b to illustrate this behaviour and to facilitate direct comparison.

### 3.1.3   $HNO_3$

A custom-made permeation source was used to provide a constant, known flow of $HNO_3$ (with $\sim 8$ % $NO_x$ impurity) to the TD-CRDS inlet. The permeation source consisted of a length ($\approx 1$ m) of PFA tubing immersed in 66 % $HNO_3$ solution held at 50 °C through which 100 sccm of dry, zero air was passed. The concentration of $HNO_3$ and, thus, its permeation rate, $(1.62 \pm 0.2) \times 10^{-4}$ sccm, was derived by measuring the optical extinction of $HNO_3$ at 185 nm using the absorption cross section of Dulitz et al. (2018). The uncertainty is related to the uncertainty in the absorption cross section and the reproducibility of the output. The $HNO_3$ thermogram (Figs. 2 and S2c) has a plateau at temperatures above $\approx 800$ °C. In the plateau region of Fig. 2, the $HNO_3$ mixing ratio measured is $13.0 \pm 0.8$ ppb, which (within combined uncertainties) is in agreement with the expected value $(15.2 \pm 1.98$ ppbv) calculated from the permeation rate and uncertainty in the dilution factor. We cannot rule out some loss of $HNO_3$ in the tubing connecting the permeation source to the TD-CRDS, although previous studies have shown that irreversible losses are $\sim 5$ % or less under dry conditions (Neuman et al., 1999). We note that inlet loss of $HNO_3$ is minimised under ambient sampling conditions, as only a short section ($\sim 20$ cm) of the quartz tubing at ambient temperature is upstream of the heated section in which $HNO_3$ is converted to $NO_2$. Therefore, our observations are in accord with previous studies that found the complete conversion of $HNO_3$ to $NO_2$ in similar set-ups (Day et al., 2002; Di Carlo et al., 2013; Wild et al., 2014; Womack et al., 2017).

### 3.1.4   $N_2O_5$

A sample of $N_2O_5$ was prepared by flowing 80 sccm of synthetic air over $N_2O_5$ crystals, kept at $-70$ °C, with further dilution with 20 slpm synthetic air. An 8.5 cm long nylon tube was used to reduce $HNO_3$ impurity. Two distinct dissociation steps can be observed in Fig. 2. The first step, between 50 and 185 °C (in which $NO_2$ increases to 4.2 ppbv, see Fig. S2d), is due to the dissociation of $N_2O_5$ to $NO_2 + NO_3$. In the second step, in which $NO_3$ is dissociated to $NO_x$ between 450 and 800 °C, the $NO_2$ mixing ratio was 9.2 ppbv. As the amount of $N_2O_5$ derived from the first dissociation step was in accord with simultaneous measurements of $N_2O_5$ using a further TD-CRDS set-up (Sobanski et al., 2016), we conclude that some $HNO_3$ was present in the sample, which was presumably the result of $N_2O_5$ hydrolysis. Our thermogram is similar to that reported by Womack et al. (2017), who also observed two steps – the first with a plateau at $T > 100$ °C and the second at $T > 650$ °C. The shift in temperature (100–150 °C) compared with our results is rationalised in Sect. 3.1.8.

### 3.1.5   HONO

Gaseous HONO was produced by flowing HCl in air (22 ppbv; relative humidity, RH, ca. 50 %), over a bed of continuously stirred sodium nitrate crystals (Wollenhaupt et al., 2000). In our set-up, the thermal dissociation of HONO to NO starts at $\sim 400$ °C and reaches a plateau (6.2 ppbv) between ca. 800 and 850 °C (Fig. 2). We did not have access to independent instrumentation to characterise the concentrations of HONO and potential impurities generated using this method. Previous investigations have reported that HONO thermally dissociates between 450 and 650 °C (Perez et al., 2007) and between 200 and 700 °C (Wild et al., 2014). The reasons for such large divergence in the positions and widths of the thermograms may be partially related to the presence of impurities in the HONO samples used, although the details of the ovens used to thermally dissociated HONO also play an important role as described in see Sect. 3.1.8.

### 3.1.6   $ClNO_2$

$ClNO_2$ was generated by passing $Cl_2$ (33 ppbv in air) over sodium nitrate at room temperature. The thermogram, depicted in Fig. 2, has two steps – one with an apparent plateau at $\sim 500$ °C and a second at $\sim 800$ °C. The lower temperature plateau in which $ClNO_2$ dissociates to $NO_2$ corresponds to that reported previously (Thaler et al., 2011; Sobanski et al., 2016; Thieser et al., 2016). The observation of further $NO_2$ formation at higher temperature is consistent with the observations of Wild et al. (2014). We hypothesise, that the second dissociation step might be associated with the presence of ClNO which dissociates to NO (and would therefore not have been detected by instruments that monitor $NO_2$ rather

than $NO_x$). Even at temperatures $> 850\,°C$, we still see an increase in the $NO_2$ signal. However, as ClNO is not considered to be an important atmospheric trace gas, this has no repercussions for the deployment of the instrument.

### 3.1.7    $NH_4NO_3$ and $NaNO_3$ particles

$NH_4NO_3$ and $NaNO_3$ particles were generated from an aqueous solution (ca. 1 g in 500 mL deionised water) using an atomiser (TSI 3076). The particles were dried prior to size selection (DMA, TSI 3080) and were diluted in a total flow of 6 slpm synthetic air which was split between a condensation particle counter (CPC) and the heated TD-CRDS inlet (after further dilution). The relative thermogram for $NH_4NO_3$ is displayed in Fig. 2 and, similar to $HNO_3$, displays a plateau region at temperatures above $830\,°C$. The shift in the thermogram when comparing $HNO_3$ and $NH_4NO_3$ (which we expect to detect in a two-step process in which $NH_4NO_3$ first decomposes to $HNO_3$) may be related to the time required to fully thermally decompose particles (e.g. of 200 nm diameter) containing several million molecules. Particle numbers (in $cm^{-3}$) detected by the CPC were converted to molar mixing ratios via the diameters and densities of the dry particles (1.72 $g\,cm^{-3}$ for $NH_4NO_3$ and 2.26 $g\,cm^{-3}$ for $NaNO_3$). The fraction of $NH_4NO_3$ detected as $NO_x$ following passage through the oven is illustrated in Fig. S5, which indicates values between $\approx 60\,\%$ and $120\,\%$ depending on particle size. The total uncertainty in the concentration was estimated as $41\,\%$, which includes $10\,\%$ uncertainty in the particle diameter (based on measured size distributions of latex calibration particles), $20\,\%$ uncertainty in the particle number (including the error in the multiple charge correction) and $10\,\%$ uncertainty in the density, due to possible differences between single crystal and bulk density. As the particle mass scales to the third order with the particle diameter, the correction for double-charged particles introduces a large uncertainty in the calculated mixing ratios, with the effect being largest in the size range between 100 and 150 nm, which probably explains the lower $NH_4NO_3$ detection efficiencies in this range. We consider the data obtained at 200 nm to be the most reliable and conclude that, similar to other TD instruments (Womack et al., 2017; Garner et al., 2020), our instrument also detects $NH_4NO_3$ particles as $NO_2$ with close to $100\,\%$ efficiency. In contrast, our experiments using $NaNO_3$ resulted in much smaller $NO_x$ concentrations despite identical experimental conditions in back-to-back experiments and produced detection efficiencies of close to $25\,\%$. While the inefficient detection of $NaNO_3$ is consistent with a previous reports suggesting that $NaNO_3$ would not be detected in TD inlets (Womack et al., 2017), it contrasts strongly with the very recent result of Garner et al. (2020), who observed quantitative conversion of $NaNO_3$ to $NO_2$ at $600\,°C$. This difference may be related to residence times in the heated section of the inlet.

### 3.1.8    Summary of thermograms

The thermograms obtained by the present instrument deviate from others reported in the literature, with the temperatures required for $50\,\%$ dissociation generally being higher by, for example, $80\,°C$ for PAN, $50\,°C$ for iPN and $150\,°C$ for $HNO_3$ respectively (Day et al., 2002; Wild et al., 2014; Sobanski et al., 2016; Thieser et al., 2016; Womack et al., 2017). This lack of agreement with other set-ups is not unexpected, as the degree of dissociation of a trace gas at any temperature depends not only on the temperature but also on the time over which the molecule is exposed to that temperature (Womack et al., 2017). To illustrate this, based on rate coefficients (related to bond-dissociation energies, BDEs) for the thermal dissociation of PAN (Bridier et al., 1991), iPN (Barker et al., 1977), $HNO_3$ (Glänzer and Troe, 1974), $N_2O_5$ (Ammann et al., 2019), $ClNO_2$ (Baulch et al., 1981) and HONO (Tsang and Herron, 1991), we calculated the theoretical $50\,\%$ conversion temperature for each molecule as a function of residence time inside the oven (see Fig. S3b). At short residence times the dependence on temperature is very steep (especially for large BDEs) which partially explains the differences between our short heated section inlet and longer ones. However, in practise, we do not know the precise average temperature of the gas at the centre of the oven nor can we characterise the axial and radial gradients in temperature in the quartz tubes; thus, the calculations of fractional dissociation (or complete thermograms) based on bond-dissociation energies are only a rough guide at best. We note that the use of different flows, oven diameters and operational pressures will strongly affect heat transfer from the oven walls to the gas; therefore, reporting the temperature of the external oven wall (as done here and in all reports in the literature) to some extent precludes comparison between different set-ups. The width of the thermograms (i.e. the temperature difference between e.g. $10\,\%$ and $90\,\%$ dissociation) will also depend on details of axial and radial temperature gradients in the tubing located within the oven as well as in the downstream section of tubing, which represents a transition regime between oven and room temperature. The impact of temperature gradients inside the quartz tube was explored by calculating the $HNO_3$ thermogram using an Arrhenius expression for its thermal dissociation and the gas residence time within the quartz tube. We initially assumed that all $HNO_3$ molecules experience the same temperature and then compared this to the situation in which $20\,\%$ of the $HNO_3$ molecules experience an $80\,°C$ lower temperature and $20\,\%$ experience an $80\,°C$ higher temperature. The resultant thermograms are displayed in Fig. S3c and indicate that the presence of temperature gradients results in an increase in the width of the thermogram from 250 to $350\,°C$.

The thermograms we report here serve only to determine the temperature needed to ensure the maximum conversion of each trace gas to $NO_2$. This is achieved in the present set-up with a temperature of $850\,°C$. Where possible, we have veri-

fied that operation at the plateau of the thermogram resulted in quantitative conversion of the traces gases and particles studied, with one exception – $NaNO_3$ particles. We further note that, in an instrument designed only to measure $NO_y$, there is no need to ensure separation (in temperature) of the thermograms for different classes of molecules.

### 3.1.9 Detection of $NH_3$

As described previously (Wild et al., 2014; Womack et al., 2017) ammonia represents a potential interference in $NO_y$ measurements. In order to quantify this interference, we measured $NO_2$ formation in air containing 131 ppbv $NH_3$ delivered by a calibrated permeation source (VICI METRONICS, permeation rate 324 ng min$^{-1}$ at 45 °C). The results are summarised in Fig. 2. In $NH_3$–air mixtures, we observe a small $NO_2$ signal, increasing slowly at first and then (from $\approx 700$ °C) rapidly with temperature; the amount of $NO_2$ observed at 850 °C corresponds to a fractional conversion of $NH_3$ to $NO_2$ of $0.006 \pm 0.002$. This result is in broad agreement with Wild et al. (2014), who found a conversion efficiency of $< 0.01$ at 700 °C. In experiments with $NH_3$ in zero air with relative humidities of 17 %, 31 % and 53 %, we were unable to observe the conversion of $NH_3$ to $NO_2$, which is again consistent with the humidity-related suppression of $NO_2$ formation observed by Wild et al. (2014).

In additional experiments, we investigated the potential influence of ozone on the $NH_3$ to $NO_2$ conversion efficiency in zero air containing $O_3$. The addition of $O_3$ results in a significant increase in $NO_2$ with a linear dependence on the $O_3$ mixing ratio (Fig. 3) and up to 11.4 % conversion of $NH_3$ to $NO_2$ at 200 ppbv $O_3$. This was not reduced measurably by the addition of water vapour to the air–$O_3$ mixture. In further experiments, we spiked air with the headspace of various organic liquids (acetone, methanol, ethanol, $\beta$-pinene, limonene and isoprene). The gas-phase mixing ratios of the organic trace gases were unknown, but the formation of $NO_2$ was suppressed or completely stopped in each case. A more quantitative investigation was carried out using a known concentration (1 ppmv gas bottle) of isoprene. We found that the addition of 30 ppbv isoprene to zero air (containing 330 ppbv $O_3$) did not significantly reduce the $NH_3$-to-$NO_2$ conversion efficiency under dry conditions, but it decreased it by a factor of 2 when the RH was increased to 50 %.

A tentative chemical mechanism, based partially on Womack et al. (2017), to explain the formation of $NO_2$ from $NH_3$ and $O_3$ at high temperatures and the processes that suppress it is given in Reactions (R11) to (R15). In this scheme, the oxidation of $NH_3$ is initiated and propagated by $O(^3P)$, which is formed from the thermal dissociation of $O_3$ (Peukert et al., 2013). This leads to the formation of NO and HNO (Reaction R13a and R13b), both of which can be oxidised to $NO_2$ (Reactions R14 and R15). Forward and reverse rate coefficients for Reaction (R11) indicate that $O_3$ is converted almost stoichiometrically to $O(^3P)$ in the

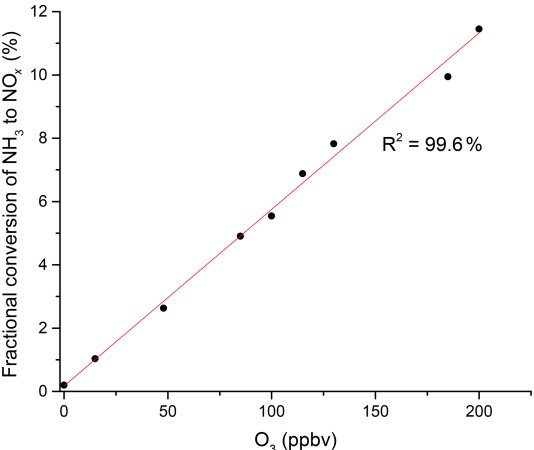

**Figure 3.** $NH_3$ to $NO_x$ conversion in the heated inlet channel of the instrument in the presence of $O_3$. The fractional conversion of $NH_3$ to $NO_x$ is calculated from the 13.1 ppmv of $NH_3$ from the permeation source.

$\approx 10$ ms reaction time in the heated inlet. The rate constants (at 1123 K) for the subsequent reactions involving $O(^3P)$ are as follows: $k_{12} = 4.4 \times 10^{-13}$ cm$^3$ molec.$^{-1}$ s$^{-1}$, $k_{13a} = 8.3 \times 10^{-12}$ cm$^3$ molec.$^{-1}$ s$^{-1}$ and $k_{13b} = 7.5 \times 10^{-11}$ cm$^3$ molec.$^{-1}$ s$^{-1}$) (Cohen and Westberg, 1991). Reaction (R12) converts 0.3 % of the initial $NH_3$ molecules to $NH_2$ within 10 ms (at 100 ppbv $O_3$ and 1123 K).

$$O_3 + M \rightarrow O(^3P) + O_2 + M \tag{R11}$$

$$O(^3P) + NH_3 \rightarrow NH_2 + OH \tag{R12}$$

$$NH_2 + O(^3P) \rightarrow NO + H_2 \tag{R13a}$$

$$\rightarrow HNO + H \tag{R13b}$$

$$HNO + O(^3P)/OH/H \rightarrow NO + OH/H_2O/H_2 \tag{R14}$$

$$NO + O(^3P) + M \rightarrow NO_2 + M \tag{R15}$$

The experimental results obtained in zero air indicate that reactions involving $O(^3P)$ from $O_3$ thermolysis can result in the conversion of $NH_3$ to NO and $NO_2$. These results could, however, not be reproduced when adding $NH_3$ to ambient air sampled from outside of the building. In this case, the addition of $NH_3$ (at 50–60 ppbv $O_3$) did not result in a measurable increase in $NO_2$, which was in accord with the observations of Womack et al. (2017). The scavenging of $NH_2$ radicals and $O(^3P)$ by both volatile organic compounds and $H_2O$ provides a likely explanation for this. Womack et al. (2017) also found that the addition of 100 ppbv CO can reduce the conversion of $NH_3$ to $NO_x$.

In summary, our experiments indicate that the conversion of $NH_3$ to $NO_2$ is suppressed in ambient air samples, or in synthetic air with added VOCs and water. The ambient air used in these experiments was from an urban and polluted environment (typical $NO_x$ levels between 10 and 50 ppbv; see

Sect. 4.2). As high levels of atmospheric $NH_3$ are associated with agricultural activity (Langford et al., 1992; Schlesinger and Hartley, 1992) and are often accompanied by high $NO_x$ and VOC levels, the $NH_3$ interference under these conditions is most likely to be small compared with ambient $NO_z$ levels. Long-term measurements of $NH_3$ have additionally found a positive correlation between $NH_3$ concentrations and ambient temperature (Yamamoto et al., 1988; Wang et al., 2015; Yao and Zhang, 2016), with the latter promoting the presence of high levels of biogenic VOCs, such as isoprene (Tingey et al., 1979), which would also help to minimise the $NH_3$-related interference.

## 3.2    Bias caused by secondary reactions in the TD ovens

Thermal dissociation techniques coupled to CRD systems for the measurement of organic nitrates suffer bias to different degrees owing to reactions of organic peroxy radicals with NO and $NO_2$ (Sobanski et al., 2016; Thieser et al., 2016). According to previous studies (Day et al., 2002; Rosen et al., 2004; Thieser et al., 2016), in experiments using iPN at an oven temperature of 450 °C, an overestimation of ANs in the presence of NO is caused by reactions of the initially formed alkoxy radical, $C_3H_7O$, which results in the formation of both $HO_2$ and $CH_3O_2$ (Reactions R16–R21).

$$C_3H_7ONO_2 + M \rightarrow C_3H_7O + NO_2 + M \qquad (R16)$$

$$C_3H_7O + O_2 \rightarrow CH_3C(O)CH_3 + HO_2 \qquad (R17)$$

$$C_3H_7O + M \rightarrow CH_3 + CH_3CHO + M \qquad (R18)$$

$$CH_3 + O_2 + M \rightarrow CH_3O_2 + M \qquad (R19)$$

$$CH_3O_2 + NO \rightarrow CH_3O + NO_2 \qquad (R20)$$

$$HO_2 + NO \rightarrow HO + NO_2 \qquad (R21)$$

$$CH_3C(O)O_2NO_2 + M \rightarrow CH_3C(O)O_2 + NO_2 + M \quad (R22a)$$

$$CH_3C(O)O_2 + NO_2 + M \rightarrow CH_3C(O)O_2NO_2 + M \quad (R22b)$$

$$CH_3C(O)O_2 + M \rightarrow CH_3CO + O_2 + M \qquad (R23a)$$

$$CH_3C(O)O_2 \rightarrow CH_2C(O)OOH \qquad (R23b)$$

$$OH + NO_2 + M \rightarrow HNO_3 + M \qquad (R24)$$

In order to investigate the potential bias in the measurement of ANs under the present experimental conditions, a set of experiments was conducted in which NO (up to 12 ppbv) was added to various amounts of iPN. The NO mixing ratio was determined by modulating the addition of $O_3$. The results (Fig. 4a) show that $NO_2$ derived from the thermal decomposition of iPN increases with the amount of NO added and results in overestimation (factor of $\sim 1.6$) at 12 ppbv NO, which is consistent with the observations by Thieser et al. (2016). This disappears when 19 ppmv ozone is added in front of the cavity so that $NO_x$ rather than $NO_2$ is measured (blue data points). This is readily explained by the compensation of the additional $NO_2$ formed via reactions of NO with $RO_2$ by an equal loss in NO, which is only detected when introducing $O_3$. This is illustrated graphically in Fig. S4.

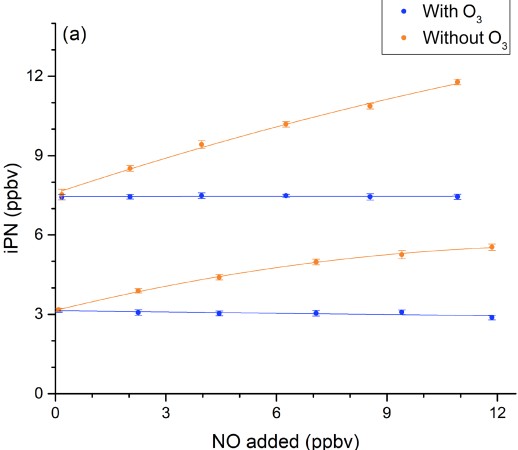

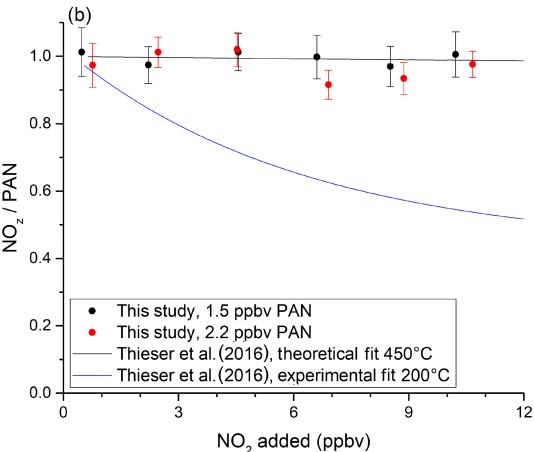

**Figure 4.** Investigation of bias caused by reactions of NO with $HO_2$ and $RO_2$ when measuring iPN. **(a)** NO varied for two initial iPN mixing ratios in the presence (blue data points) and absence (orange data points) of added $O_3$. The $NO_x$ background signal from the iPN cylinder was subtracted from the iPN mixing ratios. **(b)** The investigation of bias caused by the recombination of $RO_2$ and $NO_2$ during the thermal decomposition of PAN. In both experiments, the oven temperature was 850 °C. In both plots, the error bars indicate the standard deviation over the averaging interval.

We also explored the potential for bias caused by the recombination of $CH_3C(O)O_2$ and $NO_2$ (Reaction R22b), following the thermal decomposition of PAN (Reaction R22a). Thieser et al. (2016) reported that PAN was underestimated by a factor 0.45 when adding 10 ppbv $NO_2$ to an air sample containing PAN at 200 °C. This behaviour was not apparent at 450 °C, which is related to the decomposition (Reaction R23a) or isomerisation (Reaction R23b) of $CH_3C(O)O_2$ at the higher temperature. The results of a similar experiment with our 850 °C inlet are presented in Fig. 4b. In this plot, the measured $NO_z$ relative to the input PAN is plotted versus the mixing ratio of added $NO_2$. For PAN concentrations from 1.5 to 2.2 ppbv, no effect was observed for $NO_2$ concentrations

of up to 10 ppbv. This is consistent with the reaction scheme presented by Thieser et al. (2016) at 450 °C.

A potential source of bias when measuring $HNO_3$ includes its reformation via the reaction of OH and $NO_2$ (Reaction R24). Compared with the RO and $RO_2$ radicals formed in the thermal dissociation of PNs and ANs, OH exhibits a higher affinity for surfaces and is likely to be efficiently removed at the oven wall. Day et al. (2002) estimated that wall losses are the dominant OH sink and that the resulting underestimation of $HNO_3$ would be < 2 % for $NO_y$ levels < 5 ppbv. At our oven temperature, the diffusion coefficient for OH ($D_{OH}$) can be calculated according to Tang et al. (2014):

$$D_{OH}(1123\,K) = D_{OH}(296\,K) \cdot \frac{296}{T}^{-1.75}.$$ (2)

Using an average of the literature values for $D_{OH}$ at room temperature from Ivanov et al. (2007), 165 Torr cm$^2$ s$^{-1}$, and Bertram et al. (2001), 192 Torr cm$^2$ s$^{-1}$, a value of $D_{OH}$ (1123 K) = 1841 Torr cm$^2$ s$^{-1}$ was derived. The maximum rate constant for OH wall loss (assuming laminar flow) can subsequently be approximated according to Zasypkin et al. (1997):

$$k_{wall} = \frac{D_{OH} \cdot 3.66}{r^2 \cdot p}$$ (3)

With the radius of the oven quartz tube $r$ (0.45 cm) and the pressure $p$ (760 Torr), the maximum value of $k_{wall}$ is 44 s$^{-1}$. The first-order loss rate coefficient for the reaction of OH with $NO_2$ is given by $k_{(OH+NO_2)}$ [$NO_2$], where $k_{(OH+NO2)}$ is the rate coefficient for the reaction between OH and $NO_2$ at 1123 K $\sim 5 \times 10^{-14}$ cm$^3$ molec.$^{-1}$ s$^{-1}$ (Ammann et al., 2019) and [$NO_2$] = $6.5 \times 10^{10}$ molec. cm$^{-3}$ (the concentration of 10 ppb $NO_2$ at the pressure and temperature of the oven). The first-order loss rate of OH via reaction with $NO_2$ is then $3 \times 10^{-3}$ s$^{-1}$. Clearly, the efficiency of uptake of OH to the wall would have to be very low in order to reduce the maximum value of 44 s$^{-1}$ to values that are comparable to the reaction with $NO_2$, which is very unlikely. We conclude that reformation of $HNO_3$ via Reaction (R24) will not bias measurements of $HNO_3$ with the present set-up.

## 3.3 Denuder characterisation

The efficiency of the removal of $NO_y$ trace gases and the transmission of submicron particles was determined in a series of experiments, which are described below.

### 3.3.1 Transmission of ammonium nitrate particles (10–414 nm)

In order to characterise the transmission of the denuder for particles of different diameter, a constant flow of particles was generated by passing 3.3 slpm of nitrogen through an atomiser (TSI 3076) containing an aqueous solution of ammonium nitrate. The flow rate (3.3 slpm) was matched to the typical sampling flow through the denuder. A total of 0.28 slpm of the flow was sampled into a scanning mobility particle sizer (SMPS)/CPC system (TSI 3080 and TSI 3025A) to measure the number density and size distribution (10–414 nm) of the ammonium nitrate particles. The flow was delivered either directly to the SMPS/CPC via straight metal tubing (length 27 cm, inner diameter 0.9 cm), for which we assume 100 % particle transmission, or via the denuder. Thus, the ratio of the particle numbers in each size bin represents the size-dependent denuder transmission. As shown in Fig. 5, the transmission of the denuder is > 80 % for particles between 30 and 400 nm in diameter. As expected, some diffusive loss is observed for particles < 20 nm in diameter and loss due to impaction and/or settling is observed for particles >300 nm. The particle transmission $T$ as a function of particle diameter $D$ `CE3` can be represented by the following empirical expression:

$$T(\%) =$$
$$\frac{D(nm) - 5.79}{0.035 + 0.010 \cdot (D - 5.79) + 1.78 \cdot 10^{-6} \cdot (D - 5.79)^2}$$ (4)

The particle transmission through the denuder channels was also calculated using the Particle Loss Calculator (PLC) developed by von der Weiden et al. (2009). The results of this calculation are plotted in Fig. 5. The observed loss of particles smaller than 40 nm is not replicated by the PLC, which was developed for cylindrical piping and not for the square honeycomb shape of the denuder; thus, it does not consider losses due to impact at the finite surface area which the gas and particle flow is exposed to at the entrance to the honeycomb. The PLC does a better job at predicting a reduction in transmission for the largest particles that we measured, indicating a transmission of 74 % at 1 µm and 45 % at 2 µm. Therefore, in certain environments, nitrate associated with coarse-mode particles represents a potential bias for TD-CRDS measurements of $NO_y$.

### 3.3.2 Efficiency of removal of $NO_y$ trace gases

The efficiency of the removal of trace gases in the denuder under typical flow conditions (3.3 slpm) was investigated for NO, $NO_2$, PAN, iPN, HONO, $N_2O_5$, $ClNO_2$ and $HNO_3$ as representative $NO_y$ species. The efficiency of removal of each trace gas (generally present at 5–40 ppbv) was determined by measuring its relative concentration when flowing through the denuder (pNit channel) and when bypassing the denuder ($NO_y$ channel). The results (Fig. 6) indicate that, in dry air, all of these trace gases were removed with an efficiency of close to 100 %. However, when the main dilution flow was humidified, RH-dependent breakthrough of NO was observed, with only 60 % stripped from the gas phase at an RH close to 100 %. HONO was removed with 85 % efficiency at an RH of 46 %, and $ClNO_2$ was removed

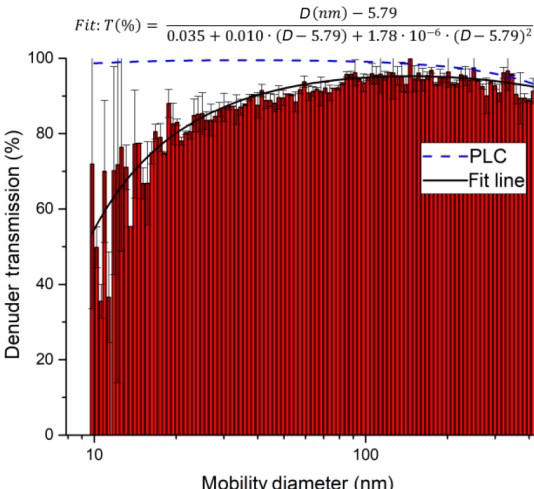

$$Fit: T(\%) = \frac{D(nm) - 5.79}{0.035 + 0.010 \cdot (D - 5.79) + 1.78 \cdot 10^{-6} \cdot (D - 5.79)^2}$$

**Figure 5.** Transmission of ammonium nitrate particles through the denuder inlet. Relative transmissions are derived by dividing the number size distribution when sampling through the denuder by a size distribution obtained without the denuder. Error bars are based on the standard deviation of three consecutive measurements with and without the denuder. An aerosol flow of 3.3 slpm was directed through the denuder (diameter 3 cm, see Sect. 2.4), and a DMA subsequently sampled 0.3 slpm from the stream exiting the denuder. The plot also includes a fit of the experimental (solid, black line) data and a theoretical transmission distribution computed with the Particle Loss Calculator (PLC).

with a 75 % efficiency at an RH of 60 %. In contrast, humidification only had a marginal effect on the scrubbing efficiency for $NO_2$, iPN and $HNO_3$, for which an efficiency of $\geq 95\%$ was observed. The precise values that the removal efficiencies in Fig. 6 were determined from are listed in Table S2.

In further experiments, we examined the potential for re-release of $NO_y$ that had previously been stored in the activated carbon substrate of the denuder. In these experiments, in which either $NO_x$ or $NO_y$ was continuously monitored, the denuder was exposed to a flow of 9.5 ppm iPN in dry nitrogen for 90 min during which $2.30 \times 10^{17}$ molecules of iPN were stripped from the gas phase and deposited onto the denuder. This exposure is equivalent to a month-long exposure to 20 ppb of iPN. The air passing through the denuder was subsequently humidified to an RH of 65 %. The results (Fig. 7a) indicate a high initial (11:40–11:50 UTC) rate of release of $NO_x$ under humid conditions (resulting in a maximum mixing ratio of 2 ppbv at 11:45 UTC). At 11:50 UTC, humidification of the air was stopped and the rate of release of $NO_x$ dropped gradually towards zero. During a second period, in which the air was again humidified (from 11:50 UTC onwards), $NO_x$ was released from the denuder, albeit at a lower rate than during the first humidification. From 12:25 UTC onwards, the oven behind the denuder was heated to 850 °C so that $NO_y$ was added. No significant increase in $NO_2$ was

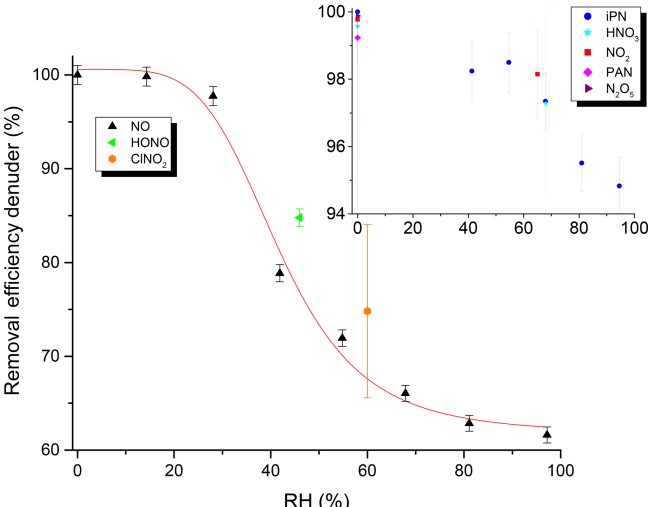

**Figure 6.** Removal efficiency of the denuder for various $NO_y$ trace gases as a function of RH. Units of the inset are identical to the main graph. See Table S2 for the exact values and information on the error determination.

observed, indicating that the trace gas(es) species released from the denuder surface upon humidification are predominantly $NO_x$. During this experiment, $2.55 \times 10^{15}$ molecules of $NO_x$ desorbed from the denuder, indicating that the major fraction of iPN molecules remained stored on the denuder surface upon humidification.

Similar denuder exposure experiments were performed with $HNO_3$ and $NO_2$. For $HNO_3$, no evidence for desorption of $NO_x$ or $NO_y$ during exposure to humidified air was observed, whereas $NO_2$ exhibited a similar behaviour to iPN (Fig. 7b). After loading the denuder with 5 sccm from a 0.831 ppm $NO_2$ gas bottle for 4.8 d (a total of $7.60 \times 10^{17}$ molecules were deposited as derived from the flow rate, the exposure time and the gas bottle mixing ratio), the effect of passing humidified air through the denuder was to release $NO_x$, which was observed at concentrations up to $\approx 39$ ppbv. While the relative humidity was kept constant at close to 100 %, the $NO_x$ released decreased over time so that after 30 min, 3.2 ppbv of $NO_x$ could still be detected. By switching the $O_3$ source off (at $\approx 10:45$ UTC), the $NO_2$ measured went to $\approx 0$, indicating that predominantly NO was released (and not $NO_2$). Integrating these results over time yielded a value of $1.63 \times 10^{16}$ molecules desorbed NO from the denuder surface in humid air. Qualitatively similar results, i.e. humidity-induced formation and release of $NO_x$ from the denuder, were observed when the denuder was exposed to variable levels of $NO_x$ (i.e. up to 20 ppbv) under dry conditions for periods of weeks.

Clearly, adsorption of water molecules onto the denuder surface can initiate and/or catalyse chemical transformation at the surface that convert stored $NO_z$ into forms which can desorb and be detected as $NO_x$. To further our understanding

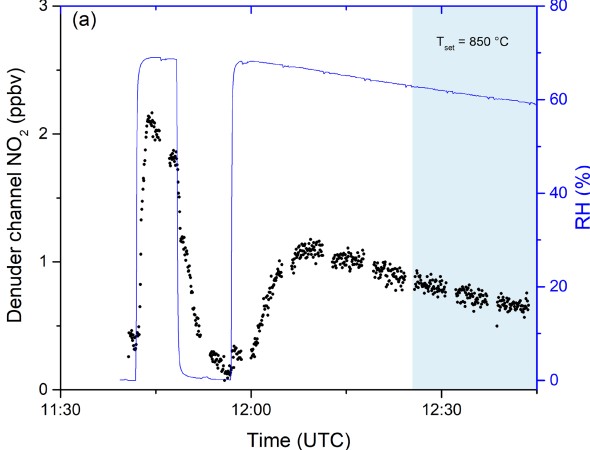

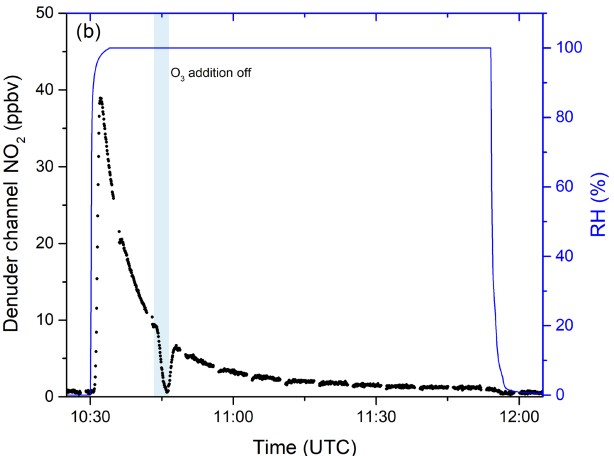

**Figure 7. (a)** Release of $NO_x$ from the denuder in humid air after exposure to 9.5 ppmv iPN for 1.5 h. Relative humidity was measured before passing through the denuder. The blue shaded area signifies the period in which the inlet oven was heated to 850 °C. Changes in RH are achieved by flowing parts of the zero air stream through deionised water. **(b)** The release of $NO_x$ from the denuder in humid air after exposure to 0.83 ppmv $NO_2$ for 4.8 d. $O_3$ addition was switched off during the blue shaded period.

of the underlying processes that occur upon humidification, a series of experiments were conducted to examine the adsorption of water on the denuder. In these experiments, the denuder was first dried by exposing it to dry air for several hours until the relative humidity of the air exiting the denuder was close to zero. Subsequently, humidified air was passed through the denuder and the RH of air exiting it was continuously monitored. The results of an experiment in which the air was humidified to 68 % are shown in Fig. 8a. After 77 min of exposure to this humidity, the RH of the air exiting the denuder acquired a maximum value of 64 %. After switching back to dry synthetic air (at 09:37 UTC), ∼ 60 minutes passed before the RH dropped to values close to zero. In this

period, the RH did not decrease monotonically, with the rate of change of relative humidity exiting the denuder revealing a number of discreet steps. Figure 8b plots the derivatives (dRH/dt) of the drying phases of a series of experiments in which the initial RH was varied between 47 % and 75 %. A similar pattern emerges for each experiment with the greatest desorption rates occurring at the beginning of the drying phase followed by a minimum in the desorption rate and a second maximum (at ≈ 15 % relative humidity). This behaviour is a clear indication that $H_2O$ is bound to more than one chemically or physically distinct surface sites on the activated carbon.

By measuring the change in the RH of the air flowing into and out of the denuder, we can derive an equilibrium adsorption isotherm for $H_2O$ at the active carbon surface. An example is given in Fig. 9 where it is also compared to a literature isotherm for the adsorption of water vapour on activated carbon fibre (Kim et al., 2008). The data of Kim et al. (2008) have been scaled by matching the number of adsorbed water molecules at an RH of 65.9 % to our observed value at an RH of 67.2 %. The exposure of carbonaceous surfaces to inert gases at high temperatures (2000 °C) reduces the capacity for water uptake, whereas functionalising the carbon surface with oxygen-containing groups (e.g. from $HNO_3$) enhances the water adsorption capacity (Dubinin et al., 1982; Barton and Koresh, 1983; Liu et al., 2017). Thus, in our experiments, the uptake of gas-phase $NO_y$ is expected to generate oxygenated sites on our denuder surface, which, in turn, will influence water uptake and subsequent further trace gas accommodation.

The chemistry leading to the formation of gas-phase $NO_x$ from $NO_y$ trace gases adsorbed at the denuder surface under humid conditions cannot be elucidated in detail with our experimental set-up. However, a strong humidity dependence in the heterogeneous generation of HONO and NO from $NO_2$ adsorbed on soot particles has been reported (Kalberer et al., 1999; Kleffmann et al., 1999). Formation of HONO from $NO_x$ has also been observed on wet aerosol and ground surfaces in field studies (Lammel and Perner, 1988; Notholt et al., 1992). Previous investigations report the adsorption of $NO_2$ on activated carbon at ambient or close to ambient temperatures (30–50 °C), followed by its reduction to NO, with the simultaneous oxidation of the carbon surface (Shirahama et al., 2002; Zhang et al., 2008; Gao et al., 2011). These results are consistent with our observation of processes such as the conversion of $NO_2$ to NO at the denuder surface under humid conditions. In our experiments, we observed that $NO_2$ was converted to NO (rather than HONO) at the denuder surface. It is possible that the initial step is the formation of HONO (e.g. by the surface-catalysed hydrolysis of $NO_2$ or iPN) which undergoes further reduction on the surface to NO. The release of HONO and NO has also been observed from soil, after the nitrification of $NH_3/NH_4^+$ or the reduction of $NO_3^-$ (Su et al., 2011; Pilegaard, 2013; Meusel et al., 2018). Oswald et al. (2013) found compara-

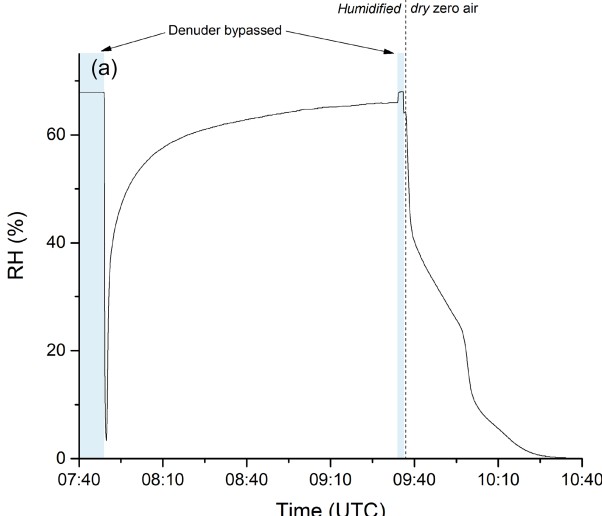

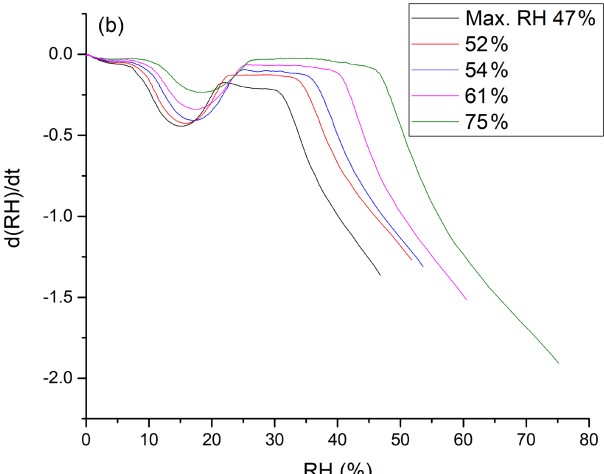

**Figure 8. (a)** RH of humidified zero air after passing through the denuder. The initial RH was determined by bypassing the denuder before and after the experiment. Zero air was humidified by flowing a fraction of the stream through deionised water stored in a glass vessel. The time at which the experiment was conducted is given on the $x$ axis. Until ca. 09:35 UTC, zero air with constant humidity (RH ca. 68 %) was sent through the denuder. Afterwards the denuder was exposed to dry zero air. **(b)** Derivative of the measured RH during the drying period. The step during the drying phase occurs in a higher RH area when starting the drying from a larger RH value.

ble HONO and NO emission fluxes from non-acidic soils, providing another example of the heterogeneous formation of HONO from other atmospheric nitrogen species, followed by the gas-phase release of NO.

We conclude that the use of this denuder type (and the assumption of complete removal of gaseous $NO_y$) may potentially result in a positive bias in measurements of particle nitrate owing to the variable breakthrough and release of $NO_x$

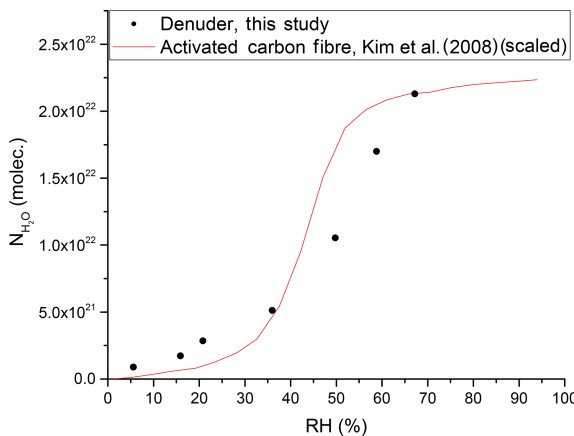

**Figure 9.** Number of adsorbed water molecules onto the denuder surface at equilibrium versus RH. The red line represents (scaled) results from a study on activated carbon fibre (Kim et al., 2008).

(dependent on the historical exposure of the denuder and relative humidity). Our findings may be applicable (at least in a qualitative sense) to similar denuders using activated carbon surfaces, and careful characterisation of the $NO_y$ components' capacity to adsorb, breakthrough and release should be carried out prior to use in the field.

Reliable surface reactivation techniques for similar denuders would be useful to ensure continuous, efficient scrubbing of $NO_y$ and $NO_x$ and circumnavigate the potential overestimation of pNit. In this regard, attempts to "reactivate" the denuder by cleaning with distilled water, drying at 50 °C and exposure to ca. 300 ppbv $O_3$ for 1 h did not result in an improvement of the direct NO breakthrough or in the background pNit signal upon humidification. Surface-sensitive spectroscopic investigation of the water-induced transformation of organic and inorganic $NO_y$ to $NO_x$ (and its subsequent release to the gas phase) on denuder surfaces would be useful in resolving these issues.

### 3.4 Comparison of the TD-CRDS with existing methods for $NO_x$ and $NO_y$ and instruments

In this section, our instrument's performance is compared to other methods for $NO_x$ and $NO_y$ detection. We consider only instruments that measure $NO_x$ and/or $NO_y$ and not those that measure individual trace gases from each family.

$NO_x$ has traditionally been measured by two-channel CLD instruments, which have one channel for NO and one channel for $NO_2$ respectively: NO is detected by chemiluminescence from its reaction with $O_3$, and $NO_2$ is converted to NO on a molybdenum converter or by photolysis at wavelengths close to 390 nm. Examples of this type of instrument as well as the reported levels of detection (LODs) and overall uncertainties are listed in Table 1. In Table 1, we also list the LOD and uncertainty of a recently developed CRDS set-up which is

**Table 1.** Comparison of $NO_x$ and/or $NO_y$ measurements.

| Species | Reference | Method | $2\sigma$ level of detection (integration time) | Uncertainty (%) |
|---|---|---|---|---|
| $NO_x$ | Parrish et al. (2004) | CLD | 20 pptv (1 s) | 10 |
| | Fuchs et al. (2009) | CRDS | 22 pptv (1 s) | 5 |
| | Wild et al. (2014) | CRDS | < 30 pptv (1 s) | 5 |
| | Reed et al. (2016) | CLD lab<br>CLD aircraft | 2.5 pptv (60 s)<br>$\sim$ 1.0 pptv (60 s) | 5<br>5 |
| | This study | CRDS | 40 pptv (60 s) | 6 |
| $NO_y$ | Fischer et al. (1997) | CLD | 200 pptv (6 s) | 25 |
| | Williams et al. (1998) | CLD "BNL"<br>CLD "NOAA" | $\sim$ 50 pptv (1 s)<br>20 pptv (1 s) | 10<br>18 |
| | Day et al. (2002) | LIF | $\sim$ 10 pptv (10 s) | < 5 |
| | Parrish et al. (2004) | CLD | 36 pptv (1 s) | 10 |
| | Wild et al. (2014) | CRDS | < 30 pptv (1 s) | 12 |
| | Pätz et al. (2006) | CLD<br>CLD | 51 pptv (1 s)<br>100 pptv (1s) | 13<br>9 |
| | This study | CRDS | 40 pptv (60 s) | 15* |

Note: * Refers to gas-phase $NO_y$ only.

similar in principal of operation to the one described here (Fuchs et al., 2009).

$NO_y$ has frequently been measured by CLD instruments with Au and/or MoO-coated thermal dissociation inlets that reduce $NO_z$ to NO, which is detected as described above for CLD-$NO_x$ instrument. Although such instruments have very low detection limits, they have been shown to be vulnerable to degradation of the $NO_y$ conversion efficiency and suffer from interferences by HCN, $CH_3CN$ and $NH_3$ (Kliner et al., 1997) as well as loss of $NO_y$ in the inlet (Zenker et al., 1998; Parrish et al., 2004). Table 1 summarises the LODs and uncertainties reported by other $NO_y$ instruments.

Table 1 indicates that the total uncertainty of the present instrument is comparable to those reported for both $NO_x$ and $NO_y$. However, our present detection limit for both $NO_x$ and $NO_y$ is worse than that reported (for $NO_2$) for the same instrument in 2016 (Thieser et al., 2016), which is a result of mirror degradation since that study.

## 4 Application of the instrument in field experiments

### 4.1 $NO_x$ intercomparison and pNit measurements during the AQABA campaign

The first deployment of the instrument was during the AQABA (Air Quality and climate change in the Arabian Basin) ship campaign in summer 2017. From 31 July to 2 September, the ship "*Kommander Iona*" followed a route from southern France via the Mediterranean Sea, the Suez Channel, the Red Sea, the Arabian Sea and the Arabian Gulf to Kuwait and back. The instrument was located in a container in front of the ship, with the inlet ovens located in an aluminium box on the roof of the container. The (unheated) tips of quartz inlet tubes protruded about 15 cm from the side of the aluminium box. Here, we compare the $NO_x$ and pNit measurements with other measurements of these parameters made during the campaign.

During AQABA, $NO_x$ levels ranged from a few parts per trillion by volume (maritime background) up to several tens of parts per billion by volume in heavily polluted air masses in shipping lanes or in harbours. In Fig. 10a, we compare $NO_x$ measured with the TD-CRDS with the results of a chemiluminescence detector (CLD 790 SR, ECO PHYSICS; Tadic et al., 2020), which measured NO and $NO_2$. The data points represent 1 min averages for the entire campaign, excluding air masses that were contaminated by the ship's exhaust. Additionally, periods with very high $NO_x$ variability were not included, with a data point being discarded whenever the differences in mean values exceeded 2 ppbv for consecutive data points.

A bivariate fit to the data sets (York, 1966), which incorporates total uncertainties for both instruments (CLD: 8.6 %; TD-CRDS: 11 % + 20 pptv × RH/100) resulted in a slope of $0.996 \pm 0.003$ and an intercept of −1.3 pptv. This very good agreement serves to underline the general applicability of the

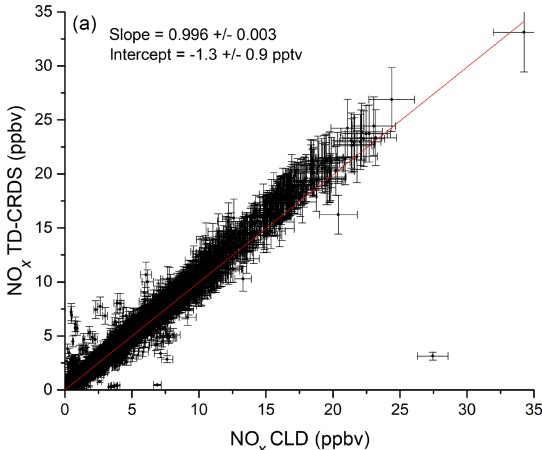

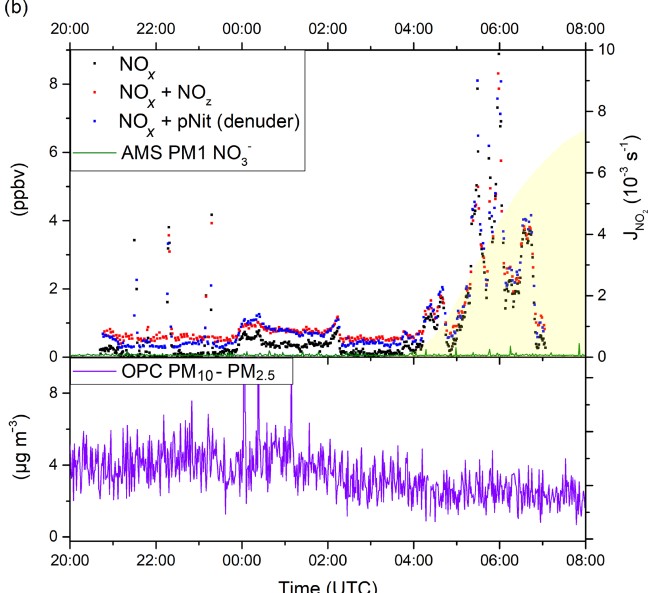

**Figure 10. (a)** Correlation between the TD-CRDS $NO_x$ measurements (1 min averages) and an independent CLD $NO_x$ instrument from the AQABA campaign. Data obtained during phases of very high $NO_x$ variability have been excluded (see Sect. 4.1). See Fig. S6 for a histogram of the $NO_x$ data points. **(b)** pNit measurements using the denuder channel (blue data points) during AQABA and comparison with particulate $NO_3^-$ from an AMS (aerosol mass spectrometer). The discrepancy towards the AMS and the correlation with the $NO_x$ mixing ratios indicate a positive bias in the pNit measurements, caused by humidity effects on the denuder surface. OPC (optical particle counter) measurements are added in the lower panel of **(b)** in order to assess the potential influence of coarse-mode aerosol nitrates.

TD-CRDS in $NO_x$ measurements, even under difficult conditions (e.g. a non-static platform). In this context, we note that the deployment on a ship resulted in a degradation in performance (the LOD was $\approx 100$ pptv) owing to the ship's motion, especially in heavy seas, which resulted in drifts in the instrument zero.

Figure 10b shows a ca. 10 h time frame with pNit measurements from the denuder channel of the TD-CRDS. Unfortunately, the TD oven of the denuder channel broke down very early in the campaign and was not operational afterwards. The data from the pNit channel are presented along with the $NO_x$ and $NO_y$ measurements as well as particulate nitrate mass concentrations measured by an aerosol mass spectrometer (Aerodyne HR-ToF-AMS; DeCarlo et al., 2006; Brooks et al., 2020). The night-to-day transition is indicated via the $NO_2$ photolysis rates $J_{NO_2}$ derived from a spectral radiometer (Metcon GmbH). The relative humidity was $> 80\%$ throughout the period shown. During the two periods when, apart from some short spikes, $NO_x$ was very low (21:10–23:45 and 02:20–03:40 UTC), TD-CRDS data indicate the presence of 300–400 pptv of pNit, which would then constitute $\sim 80\%$ $NO_y$. Such mixing ratios of particulate nitrate are not commensurate with those measured by the AMS, which, on average, are a factor of 6–8 lower. As the AMS does not detect particles larger than $\sim 600$ nm with high efficiency (Drewnick et al., 2005), the difference could potentially indicate that a significant fraction of the particulate nitrate is associated with coarse-mode aerosol. In the lower panel of Fig. 10b, we plot the coarse-mode aerosol mass concentration determined from measurements of an optical particle counter (OPC) that measures particles between 0.2 and 20 μm. In the two low-$NO_x$ periods outlined above, the OPC-derived aerosol mass concentrations were between 3 and 5 μg m$^{-3}$. If 10% of this coarse-mode aerosol mass concentration was nitrate, which is a typical value in the Mediterranean (Koulouri et al., 2008; Calzolai et al., 2015; Malaguti et al., 2015), this would account for 100–200 pptv of the pNit observed by the TD-CRDS and not by the AMS. However, the time profile of pNit measured by the TD-CRDS is not consistent with those of either the OPC or the AMS and rather resembles the $NO_x$ mixing ratios. This strongly suggests that the large difference between pNit reported by the TD-CRDS and the AMS does not result from the non-detection or detection of supermicron particulate nitrate by the AMS but instead results from the denuder artefacts described in Sect. 3.3.2. This short case study serves to highlight the potential positive bias in denuder-based TD-CRDS measurements of pNit under humid field conditions.

## 4.2 Ambient $NO_x$ and $NO_y$ measurements in an urban environment (Mainz, Germany)

$NO_x$ and $NO_y$ mixing ratios were obtained in air sampled outside the Max Planck Institute for Chemistry (MPIC). The MPIC (49°59′27.5″ N 8°13′44.4″ E) is located on the outskirts of Mainz but within 200 m of two busy two- and four-lane roads and within 500 m of additional university buildings as well as commercial and residential areas. The city of Mainz (217 000 inhabitants) is located in the densely populated Rhine–Main area along with the cities of Frankfurt (753 000 inhabitants) and Wiesbaden (278 000 inhabitants);

the air in this region is strongly influenced by local pollution. The sampling location was on the top floor of a three-story building (ca. 12 m above ground level). Air was subsampled to the inlets of the instrument from a ∼ 1 m long 0.5 inch outer diameter PFA tube that was connected to a membrane pump and flow controller to generate a 20 slpm bypass flow. Aerosol transmission was probably < 100 % in these measurements.

Figure 11a summarises the 8 d of measurement (data coverage 82 %) as a time series for NO$_x$, NO$_y$, NO$_z$, (10 min averages) wind speed (1 h averages) and the NO$_z$/NO$_y$ ratio. The NO$_x$ and NO$_y$ mixing ratios were highly variable throughout this period, with NO$_x$ mixing ratios between 0.7 and 148.3 ppbv (mean and median values of 22.1 and 6.9 ppbv respectively). Traffic-related morning rush-hour peaks in NO$_x$ were observed on all weekdays (14, 16, 17 and 20 January) between 05:00 and 10:00 UTC. The morning NO$_x$ peak is reduced or absent on the weekends (18 and 19 January). NO$_x$ levels stayed above 50 ppbv for nearly a full day from 18:00 UTC on 16 January until 18:00 UTC on 17 January, which coincides with constantly low wind speeds and the sampling of air masses that were predominantly local, and thus highly polluted. NO$_z$ mixing ratios were usually between 0.5 and 2.5 ppbv (minimum < LOD, maximum 3.1 ppbv, mean 1.0 ppbv and median 0.9 ppbv), with NO$_z$/NO$_y$ ratios below 0.5. These values indicate that the air masses had been impacted by recent (local) NO$_x$ emissions.

The NO$_z$/NO$_y$ ratio can be used as indicator of the degree of chemical processing of an air mass. In Fig. 11b, a median diel profile (including all measurement days) for the NO$_z$/NO$_y$ ratio from the ambient measurement is shown. The diel profile displays two distinct minima in NO$_z$/NO$_y$ during the morning and evening rush hours, where NO$_z$ only makes up 5 %–10 % of the total NO$_y$. This fraction increases up to 15 % during midday and up to 25 % during nighttime, when emissions of NO$_x$ are reduced. The diel profiles of NO$_z$/NO$_y$ are strongly influenced by fresh emissions of NO$_x$. As the measurement location is strongly influenced by traffic, there is a decrease in NO$_x$ (and an increase in NO$_z$/NO$_y$) at night-time. Night-time increases in NO$_z$ (13–14, 15–16, 18–19 and 19–20 January 2020) may also be partially caused by the formation of N$_2$O$_5$ as previously observed (Schuster et al., 2009) and which would have been favoured by the low night-time temperatures (< 10 °C) in winter.

These measurements serve to illustrate the applicability of our TD-CRDS over a wide range of NO$_x$ and NO$_y$ concentrations under realistic field conditions and in the investigation of processes that transform NO$_x$ into its gas- and particle-phase reservoirs.

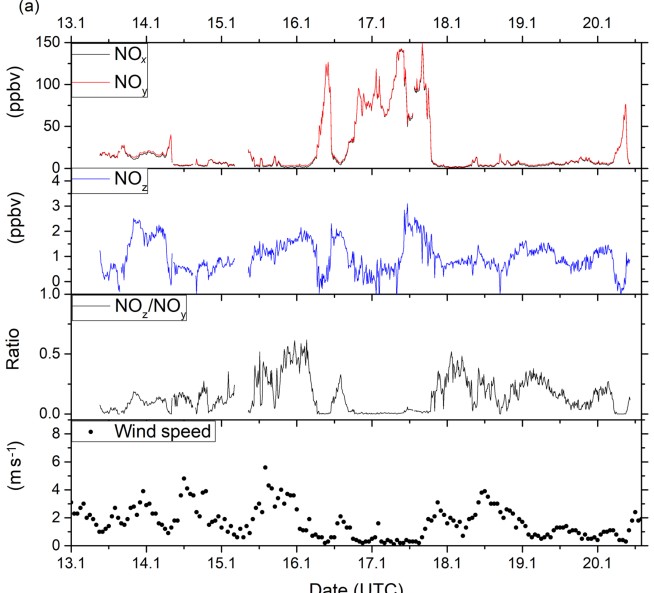

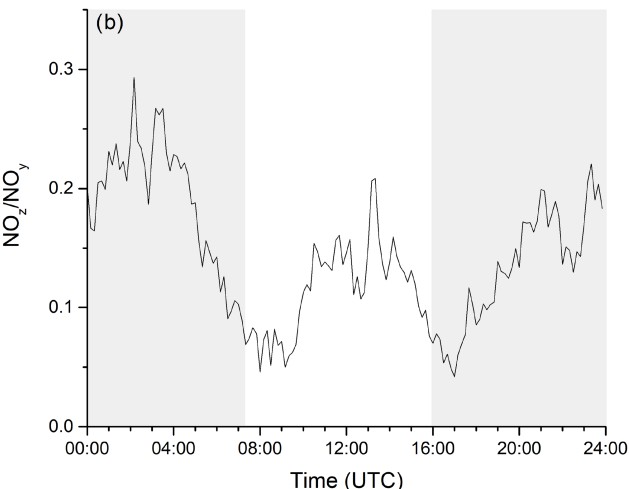

**Figure 11. (a)** Time series of NO$_x$, NO$_y$, NO$_z$, NO$_z$/NO$_y$ and wind speed from ambient measurements in Mainz, Germany, in January 2020. Highly variable NO$_x$ (between 0 and 150 ppbv) and moderate NO$_z$ (between 0 and 3 ppbv) mixing ratios were observed, identifying the sampled air masses as being dominated by anthropogenic emissions. Wind speed data was obtained from Agrarmeteorologie Rheinland-Pfalz (http://www.wetter.rlp.de TS6). **(b)** Diel profile of the NO$_z$/NO$_y$ ratio including all measurement days, showing distinct minima during the morning and evening rush hours. Shaded areas signify the time between sunset and sunrise.

## 5 Conclusions

We report on the development, characterisation and first deployment of a TD-CRDS instrument for the measurement of NO$_x$, NO$_y$, NO$_z$ and pNit. Our laboratory experiments suggest that the different gas-phase NO$_z$ species investigated

(PAN, iPN, $N_2O_5$, HONO, $ClNO_2$ and $HNO_3$) are converted with near-stoichiometric efficiency to $NO_x$ at a nominal oven temperature of 850 °C. $NH_4NO_3$ particles of 200 nm in diameter are also detected quantitatively as $NO_x$, whereas the efficiency of detection of $NaNO_3$ particles of similar diameter was closer to 25 %. The efficiency of the detection of coarse-mode particles will be further reduced by their lower transmission through the denuder.

The potential for $NH_3$ to bias $NO_y$ measurements was assessed and was found to be insignificant in ambient air or synthetic air containing VOCs and water. The conversion to $NO_2$ (by reaction with $O_3$) of atmospheric NO, as well as NO formed in the heated inlet, circumvents bias resulting from $O_3$ thermolysis (leading to an $NO_2$ overestimation) and secondary processes, which are initiated by the thermal dissociation of organic nitrates.

For our activated carbon denuder, we observed $> 90\%$ transmission for ammonium nitrate particles with diameters between 40 and 400 nm. Under humid conditions, the denuder suffered from the direct breakthrough of NO and the re-release of previously stored iPN and $NO_2$ in the form of NO, indicating a potential bias of pNit measurements using this technique and potentially limiting its deployment to low-$NO_x$ and low-$NO_z$ environments. When using comparable denuders, we recommend regular checks with humidified zero air to characterise potential breakthrough. Our experiments demonstrated that the release of $NO_x$ from the denuder exposed to humid zero air for several hours can decrease to values below 1 ppbv, which, in a first approximation, could be treated as an offset. Cycling between multiple denuders would help to reduce the size of any bias.

The performance of the instrument under field conditions was demonstrated by measurements in Mainz, Germany, and during the AQABA ship campaign. $NO_x$ measurements with the new instrument were in good agreement with those from an established, independent CLD-based instrument.

*Data availability.* The $NO_x$ data set from the AQABA campaign is available at https://doi.org/10.5281/zenodo.4134659 (Friedrich, 2020). Data used to generate the figures in this paper can be obtained upon request from the corresponding author. CE4

*Supplement.* The supplement related to this article is available online at: https://doi.org/10.5194/amt-13-1-2020-supplement.

*Author contributions.* NF developed the TD-CRDS, performed all laboratory and campaign measurements, evaluated the data sets and wrote the paper. IT and HF provided the AQABA CLD $NO_x$ measurements. JS designed the heated inlet system and performed actinic flux measurements during AQABA. JB, ED and FD provided AMS and OPC measurements from AQABA. JL and JNC designed and supervised the study and the campaigns. JNC, JL and FD contributed to the paper.

*Competing interests.* The authors declare that they have no conflict of interest.

*Acknowledgements.* We thank Ezra Wood for providing us with the activated carbon denuder and Chemours for the provision of the FEP sample (FEPD 121) used to coat the cavity walls. This work was supported by the Max Planck Graduate Center with the Johannes Gutenberg-Universität Mainz (MPGC).

*Financial support.* The article processing charges for this open-access publication were covered by the Max Planck Society.

*Review statement.* This paper was edited by Dwayne Heard and reviewed by three anonymous referees.

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

**Remarks from the language copy-editor**

CE1    Is is unclear what is located at the front of the inlet. Please clarify. Thank you.

CE2    Please note the addition.

CE3    Please check that $d$ and $D$ are now correct throughout the paper. If not, please specify exactly where changes are required. Thank you.

CE4    Please note the slight edits.

**Remarks from the typesetter**

TS1    According to our standards, changes like this must first be approved by the editor, as data have already been reviewed, discussed and approved. Please provide a detailed explanation for those changes that can be forwarded to the editor. Please note that this entire process will be available online after publication. Upon approval, we will make the appropriate changes. Thank you for your understanding.

TS2    Please see previous remark regarding editor's approval.

TS3    Please see previous remark regarding editor's approval.

TS4    Please see previous remark regarding editor's approval.

TS5    Please indicate what needs to be corrected here. "downstream" was only highlighted.

TS6    Please provide date of last access.

TS7    Please provide volume number.

TS8    Please check DOI ("not found").

TS9    Please confirm author name (as published in ACP).

TS10    Please provide exact date.

TS11    Please note that this reference must be cited as published in ACP.