# Peer review of "Measurement of $NO_x$ and $NO_y$ with a thermal dissociation cavity ringdown spectrometer (TD-CRDS): Instrument characterisation and first deployment."

_Atmospheric Measurement Techniques, 2020_

## Referee Comment (RC1) · Anonymous Referee #1 · 24 Jun 2020

This paper reports the development of an instrument to measure NO2, total NOy and total particulate nitrate based on Cavity Ring-Down spectroscopy for measuring NO2 and thermal dissociation for NOy. The accurate measurement of NOx down to low (ppt) levels is crucial for understanding the chemistry of remote atmosphere and combining such an instrument with thermal dissociation to measure total reactive nitrogen compounds and particulate nitrate further adds to the potential uses of such an instrument. Whilst a few examples of this type of approach exist in the literature, CRDS is a relatively new method and so work like this is important.

[Figure]

In general the paper details a comprehensive laboratory study of the instrument, including the thermal decomposition of different NOy species and the performance of the denuder for the particulate measurements. It is well written, easy to follow and within scope of the journal. I recommend publication subject to the following, largely minor amendments and additions.

A detailed description of the CRDS NO2 / NOx instrument is given in a previous paper (Thieser et al 2016), however I feel this new paper would benefit from some more details on the performance of the instrument to NO2 and NO. There is no mention of how these species are calibrated, or what the precision / accuracy are. Whilst this data may be able to be found elsewhere, I believe the authors should include it here as well. It would greatly assist readers wishing to get a full understanding of the performance of the instrument. I would suggest at least adding what calibration gases were used for NO and NO2, what is the accuracy and precision of these measurements at various time resolutions and what is the magnitude of any interferences.

On page 8 lines 17-20 it is stated that complete conversion of HNO3 to NO2 occurs at temperatures above 800oC, but then that the amount of NO2 detected of 13ppb is 85% of that expected based on the permeation and dilution flows. These two things do not seem to be consistent with each other – could the authors please clarify? Also, no mention is made of any potential losses of HNO3 to the surface of the instrument or the inlet, something that is often a problem with this type of instrument?

Could the authors comment on if there would be an effect of HONO on the NOy channel?

In section 3.3.1 could the authors make some comment as to how much particles greater than 414nm in diameter are transmitted? I would have thought that, especially in remote marine environments, particulate nitrate have a significant fraction on larger particles and thus provide an interference to the instrument.

In section 3.3.2 could the authors comment on how the efficiency of the denuder
changes with age and how often it may need to be regenerate or replaced.

In section 4.1 the authors state that they do not present an analysis of the NOz data s it will be presented in a future publication. I think they should at least comment on the NOz data observed. This paper is about the development of an instrument to measure NOz and to not comment on the measurements made even a little seems very strange.

Likewise section 4.2 would also benefit from an expanded discussion of the NOz data. For instance why is the diurnal cycle observed as it is, especially the nighttime peak values.

Finally I wonder if the authors could comment on now the particulate nitrate measurement could be improved. There are some suggestions given in section 3 but I think there should be something in the conclusions about this. Currently I read the paper like there was not much hope that the technique could be used for accurate particulate nitrate measurements but I am sure this is not the case, thus the authors should say so.

Page 13 line 6: 'humidified significant' does not make sense.
* * *

---

## Referee Comment (RC2) · Anonymous Referee #2 · 9 Jul 2020

Friedrich et al. describe a 2-channel thermal dissociation cavity ring-down spectrometer (TD-CRDS) for quantification of NOx, NOy, and particulate nitrate (pNit). Thermograms of peroxyacyl nitrate (PAN), isopropyl nitrate and nitric acid are presented. The potential interference from NH3 and secondary radical chemistry is evaluated. The use of an activated carbon denuder to suppress gas-phase components of NOy and transmit pNit is described. Several experiments are presented to characterize the performance of this denuder: the transmission of ammonium nitrate particles in the 10 nm - 414 nm size range and the partial removal of several trace gases (NO, NO2, PAN, iPN,

and HNO3) as a function of relative humidity (RH) which revealed inlet memory effects. Sample ambient air measurements from the 2017 AQABA campaign and ambient air measurements in Mainz, Germany are presented.

This research group has described similar instruments previously (Sobanski et al., 2016; Thieser et al., 2016) and in this paper extends the earlier measurement capabilities to now quantify NOx, NOy, and aerosol nitrate. The measurement of NOx and NOy (including NH4NO3) has previously been demonstrated by Fuchs et al. (2009) and Wild et al. (2014), so that the main novelty of this work is the use of the denuder to selectively quantify ammonium nitrate. A denuder was previously used by the Cohen group (Rollins et al., 2010) to quantify organic nitrates which dissociate at a lower inlet temperature than NH4NO3.

The measurement of NOx by CRDS through the addition of O3 is convincing. However, the paper will require considerable revision before it can be considered for acceptance. I have summarized my major concerns below. Most importantly, I am not convinced that the instrument presented here yields accurate NOy and particulate nitrate data.

Major comments

(1) Thermograms and temperature dependence of thermal dissociation:

(a) The thermograms presented in this paper are inconsistent with literature, but no convincing rationale is provided as to why that would be. (Day et al., 2002; Wild et al., 2014) have shown that thermograms of iPN and HNO3 are considerably offset from each other, which is what one would expect from their Arrhenius parameters (and is shown in the SI of this paper as Figure S2D). The "rather short" heated section of the inlet likely broadens the thermograms which is undesirable but should not have resulted in their complete overlap.

(b) The thermogram for PAN is inconsistent with a large body of literature including work by the Mainz group (Phillips et al., 2013) which showed full dissociation of PAN at an

inlet temperature of ∼150 °C (not at ∼400 °C). If the higher dissociation temperatures are a consequence of the short heater residence time, then this would be design flaw as the higher temperatures enable unwanted side reactions, increase thermal gradients, and reduce selectivity.

(c) Have the authors independently confirmed the identity, purity and concentrations of the gases they sampled by CIMS (PAN, HNO3) or GC (iPN)? For PAN, Figure S2a shows a step after ∼300 °C which may be due to the presence of an alkyl nitrate. For iPN and HNO3 Figure S2 shows "expected values" based on permeation and dilution flows, though I consider these methods reliable standards.

(d) The thermogram of NH4NO3 should be shown.

(e) N2O5, ClNO2, and HONO are important NOy components; their thermal dissociation should have been evaluated since the claim is made that the inlet's behaviour is different from that of Wild et al. (2014).

(2) Denuder performance:

(a) The denuder partially transmits gases but at a rate that appears to be dependent on environmental factors such as RH (Figure 6) and possible also ambient air temperature (not examined). The transmission of many potentially important components of NOz (e.g., N2O5, ClNO2, HONO) was not evaluated but should have been.

(b) Memory effects. The high-concentration experiments described on pg 13 are a poor way of assessing memory effects because there could be a limited number of surface sites that are overloaded when sampling a high concentration. A more realistic experiment would be to sample a low concentration of NOx for a longer time period, and then examine what comes off the denuder while sampling zero air.

(c) NOx and the aforementioned components of NOz usually dominate NOy, such that any transmission of these species constitutes a sizeable measurement error for pNit. This should have been taken into account in an explicit error analysis as part of the

pNit data reduction.

(3) Aerosol nitrate can be present on mineral dust as organic nitrates or on sea salt aerosol. Do other nitrate salts convert to NO2 in this inlet?

(4) How does the instrument perform above the 1-micron size range? The latter dominates aerosol mass in many regions.

(5) Ambient air measurements of pNit:

(a) How was the error due to break-through of NOx considered?

(b) How were denuder memory effects taken into account in the reduction of the field data?

(c) The RH dependence of denuder performance is a considerable issue considering the inlet is periodically flooded with dry zero air and then re-exposed to humid ambient air. How does this back-and-forth affect the field data?

(d) How long was the inlet for the NOy channel? Was conductive tubing used (or were aerosol lost on the inlet)?

(e) Since the TD-CRDS data were not compared with an independent measurement of NOy (Mo converter CL) or of total aerosol nitrate (High-volume impactors, PILS, MARGA or similar), the claim that this instrument accurately quantifies NOy and particulate nitrate in ambient air is not substantiated and the conclusions need to be weakened accordingly.

(6) The authors show several pages on the interference due to gas-phase NH3 only to conclude it to be insignificant. However, much more NH3 is typically present in the form of aerosol ammonium, which would evaporate in the NOy inlet. The authors should also consider and examine conversion of (NH4)2SO4 especially in the presence of O3.

(7) General organization:

(a) The introduction with its lengthy discussion of nitrogen oxide chemistry misses the mark (see detailed comment below).

(b) A critical comparison of the performance of this new instrument to existing methods (in terms of detection limits, selectivity, instrument including heater designs, etc.) is lacking and should be provided.

Specific comments

pg 1 line 10 "detection" replace with quantification

pg 1 line 14 "detection limits" Please state the level of confidence. Why does the NOy channel have half of the LOD of the NOx channel?

pg 1 line 17, 18, 19 "significant interferences", "high particle transmission", "essentially complete removal" Please be quantitative.

pg 1 line 19 "denuder suffered from NOx breakthrough" Does the breakthrough of NOx not imply that the pNit measurement is inaccurate and does not work?

pg 1 line 21-22 "NOx measurements obtained from a ship sailing through the Red Sea, Indian Ocean and Arabian Gulf .... were in excellent agreement with those taken by a chemiluminescence detector of NO and NO2." What about the NOy and pNit data during this cruise?

pg 1 line 23 "A dataset exploring variations in the NOz to NOy ratio (maximum value of 0.6) of air in a region (Mainz, Germany) with strong urban influence was measured over a one-week period in winter." and what was the conclusion?

pg 2 line 1 - pg 3 line 21 Section 1.1 "Atmospheric NOx and NOy". This paper is about a new instrument. The lengthy description of NOx and NOy chemistry and all those reactions (in particular R6-R10) are not needed and could / should be replaced by citations to the authors' own papers.

If this section stays, please fix R8. RH is already used as relative humidity and implies
a saturated alkane, with which NO3 barely reacts and usually does not form an alkyl nitrate.

pg 4 line 17 "Pyrolysis". Thermolysis or thermal dissociation are more appropriate here.

pg 4 line 25. "which overcome these limitations" Please be more specific here. One of the limitations discussed in the preceding paragraph mentions secondary chemistry by O-atoms formed from the decomposition of ozone, which wasn't addressed in this paper.

Since some of the co-authors have described multi-channel TD-CRDS instruments previously (Sobanski et al., 2016; Thieser et al., 2016), please add a short statement explaining how this new instrument differs from the old ones and what parts constitutes novelty (use of a denuder to quantify pNit).

pg 5 Section 2.1 "CRDS Operation Principals" How much of this section is duplication of (Sobanski et al., 2016; Thieser et al., 2016)? Please condense and focus on what has been changed since the earlier versions, and why.

pg 5 line 5 spell out / define STD

pg 5 line 21 Please give an uncertainty for l/d.

pg 5 line 23 "aerosol particles" should simply be particles.

Does a 2 $\mu$m pore size filter truly removes all particles? In our experience, they do not, but they remove the size range that would interfere optically. Consider rephrasing.

pg 5 line 27 "depending on "flow, pressure and inlet set-up" State typical pressures and flows. Is the inlet described anywhere? Please call out the relevant section if it is.

pg 5 line 29 300 pptv NO2 equivalent and 6.5 Torr pressure difference seem like a lot. It seems to be caused by the peculiar addition point of the zero air between the valve/denuder/inlet converter and CRDS cells and probably could be avoided altogether if zero air were added at the tip of the inlet (with larger fittings). Please provide a rationale why zero air was added this way (state advantages and disadvantages).

pg 6 line 9. "The maximum concentration of NO2 (and thus optimal conversion of NO to NO2)". Please state what fraction of NO that is converted and if the NO data were corrected accordingly.

pg 6 Section 2.3 How is temperature measured in these furnaces? Is the oven temperature identical to the temperature of the gas travelling through it?

sections 2.3 and 2.4. A critical parameter is the sample flow rate, which should be stated here.

pg 6 line 20. How short is "short"?

An experiment of the inlet setups during the two field campaigns should be provided.

pg 6 line 22. Describe the valve (make & size etc.). Are there memory effects? Pressure drops?

pg 7 line 11 "Results and discussion, 3.1 The fractional conversion of NOz to NO2 in the TD-inlets was investigated in a series of experiments in which constant flows of (separately) PAN, isopropyl nitrate and nitric acid were passed through the heated-inlet (bypassing the denuder) while the temperature was varied and NO2 was monitored"

This section describes how the experiment was conducted and should be moved to the experimental section, and not appear under "Results and discussion"

pg 7 line 20. ∼400 °C to dissociate PAN is very high. How and where exactly is this temperature measured? What is the difference between the measured temperature and the temperature of the gas stream?

pg 7 line 22 "We conclude that PAN is stoichiometrically converted to NO2" Just because the curve has flattened does not imply stoichiometric conversion. Has this statement been verified, e.g., by comparison to a CL NOy instrument or PAN-CIMS?

[Figure]

pg 7 line 24 it is also higher than Day et al. (2002), Paul et al. (2010), Di Carlo et al. (2013) and Sadanaga et al. (2016) and inconsistent with CIMS inlet performance (Slusher et al., 2004; Zheng et al., 2011; Mielke and Osthoff, 2012; Phillips et al., 2013).

pg 7 sections "3.1.1. PAN" and 3.1.2 "Isopropyl nitrate". Please comment on the possibility that the PAN and iPN sources contain impurities. PAN, for instance, will slowly decompose and form nitric acid and an alkyl nitrate.

pg 7/8 "3.1.2 Isopropyl nitrate" and "3.1.3" I am skeptical about the accuracy of these thermograms. The overlap of iPN and HNO3 is inconsistent with literature.

pg 8 line 3 "expected" Expected how?

pg 8 line 20. Please also compare with Di Carlo et al. (2013).

pg 8 lines 22-25. If the temperatures are truly that inaccurate, please consider at least rough-calibrating the temperature scale.

pg 8 line 26. "Rather short" Why so short?

In such a short inlet, the gas stream is likely heated very unevenly, leading to considerable broadening of the TD profiles. Was this intentional, or is this a design flaw? Note the broadening would not explain the overlap of iPN and HNO3 thermograms.

pg 8 line 29 if this calculation can be performed for 50% conversion, it can also be done for 10%, 20%, 30%, etc. to construct an expected TD profile, which in all likelihood will be inconsistent with Figure 2.

pg 8 line 32 "to ensure complete conversion of each trace gas to NO2". Replace complete with maximum conversion.

pg 9 line 4 "verify quantitative ... conversion to NO2". Quantitative conversion was not demonstrated in this work; In fact, Figure S2 suggests incomplete conversion ($\sim$85%; or 13/15 for HNO3).

pg 9 line 6. Section "3.1.4 NH3" Womack et al. (2017) showed NH3 to be non-issue under most conditions, which is the same conclusion reached here. Does this insignificant interference really warrant two full pages? Consider condensing.

If NH3 can be converted to NOx, what happens when NH4NO3 or (NH4)2SO4 aerosol are sampled?

pg 9 line 22 "addition of 30 ppbv isoprene to zero-air did not significantly reduce the NH3-to-NO2 conversion efficiency under dry conditions, but reduced it by a factor of two when the RH was increased to 50%" In the preceding section, it was shown that NH3 converts when there is a lot of O3 present; please clarify if isoprene was added in the absence or presence of O3 (with which it would react)?

pg 11 R22. How much CH3C(O)O2NO2 are expected in a heated inlet? Its dissociation reaction is missing.

pg 11 equation (2). Is this equation valid for the likely turbulent flow conditions in the inlet?

pg 12 line 24/26 "Particle loss calculator (PCL)" Should this be PLC?

pg 12 line 26 "which was developed for cylindrical piping and not the square honeycomb shape of the denuder and also does not take into account losses due to impact at the finite surface area which the gas/particle flow is exposed to at the entrance to the honeycomb" Based on this statement, wouldn't it be reasonable to conclude that this calculator should not be used?

pg 13 line 6. "close to 100%". Please provide a table with the precise values and some statistics.

"humidified significant" Grammar.

"RH-dependent breakthrough". During zeroing, dry air is added and some of it travels through the denuder; does the denuder require some conditioning then after the switch

back to ambient air sampling?

pg 13 line 9. "≥95%. Please give precise values. Since NOx is usually the major component of NOy, the partial and variable (as a function of RH) transmission of NOx introduces a major bias when quantifying aerosol nitrate.

pg 13 line 12. Why dry nitrogen (if the behaviour is different at high RH)? Is this really equivalent to 1 month of sampling ambient air? How was 2.30ïĆť1017 molecules derived at?

pg 13 line 20 "2.55x1015" what are the units here?

pg 13 line 23 "After loading the denuder with 5 sccm from a 0.831 ppm NO2 gas bottle for 4.8 days" NO2 cylinders usually co-emit HONO, HNO3 and NO. Was this considered?

pg 13 line 24 "(a total of 7.60 x 1017 molecules deposited)" How this value determined?

pg 15 line 14 sections 4.1 and 4.2 are not convincing since there is no independent measure of what to expect for pNit.

pg 16 line 10 Does the AMS quantify supermicron particles at all? A citation is needed.

pg 17 line 18 "> 90% transmission for ammonium nitrate". Is this statement justified when the transmission varies with size as shown in Figure 5?

pg 21 line 45. The accepted paper should be cited.

pg 22 line 28. The accepted paper should be cited.

pg 23 line 13 The paper by Womack et al. (2017) has been accepted and should be cited and not its discussion paper.

pg 24 Figure 1. Please add more detail such as dimensions; for example, indicate the diameter of the critical orifice and air pressure, and show l and d.

Figure 1 shows that ambient air is drawn in through two valves ("V")? Are these described anywhere? How much of a pressure drop do they give? What is the internal surface made of (Teflon?)?

In Figure 1, "1/2 in PFA" should be in metric units; indicate if this is outer or inner diameter and how long this section is.

In Figure 1, what does the green line represent? A chamber wall?

The Figure is inconsistent with the text as it shows a TD oven at 850 °C which is roughly the same length as the denuder; in the text, the section actually heated is described as rather short or 3 cm long, and the denuder length is stated at 10 cm.

pg 25 Figure 2. The PAN and iPN thermograms are inconsistent with literature. The caption should state the level of O3 present for the NH3 experiment.

pg 26 Figure 3 is not essential to this paper and probably should be in the supplemental.

pg 28 Figure 5. Please specify the type of aerosol diameter (geometric vs. mobility). State value of key parameters (flow rate, denuder diameter). Correct inconsistency between PLC and PCL.

pg 29 Figure 6. Please zoom in to ∼60% to 100% for NO and to 95% to 100% for the other gases. Often, NO and NO2 are the largest components of NOy. How accurate is the measurement of NOz species if ∼5% of NO2 and ∼35% of NO break through the denuder? How variable are the numbers shown in Figure 6 (add error bars)?

pg 30 Figures 7. What do the blue shades represent and why does the RH change (state in caption). As stated in the comments above, I am not convinced this experiment provides relevant information for an ambient air measurement (conc. are simply too high).

pg 31 The presentation of Figure 8 is unclear. How this experiment conducted? Is synthetic air equal to zero air (and why is it humid then)? In panel (a), why is there

a "bump" at 9:50? In panel (b), what does the derivative mean to the reader? What time? Why are there lines for different RH?

pg 32 Figure 9. Properly cite Kim et al. in the bibliography, and do not provide the full reference in the caption.

pg 33 Figure 10a. Is the slope correct? There seem to be many points above the line. Is the slope affected by outliers? In any case, it's great that NOx data agree, but since the focus of this paper is mainly on measurement of NOy, NOz and particle nitrate, a more relevant plot would be TD-CRDS NOy vs. CLD NOy, TD-CRDS NOz vs. NOz measured by other techniques, as well TD-CRDS nitrate vs PILS or MARGA nitrate.

pg 33 Figure 10b I am not sure the OPC data add anything of value here since the nitrate fraction could be changing.

Supplemental:

Figure S1. Please provide the mechanism used in the box model.

Figure S2, caption. "total uncertainty of the TD-CRDS measurements" - what is meant by this?

Figure S2D is inconsistent with Figure 2 which shows the entire TD profile of iPN overlapping with that of HNO3 - which is not observed in Figure S2D under any condition.

Figure S3. A strange and confusing way to present data.

The simulations consider oxidation of NO2 by O3 to form NO3 and subsequent formation of N2O5. Were NO3 and N2O5 sinks been considered? Please provide the full mechanism.

References cited:

Day, D. A., Wooldridge, P. J., Dillon, M. B., Thornton, J. A., and Cohen, R. C.: A thermal dissociation laser-induced fluorescence instrument for in situ detection of

[Figure]

NO2, peroxy nitrates, alkyl nitrates, and HNO3, J. Geophys. Res., 107, D6, 4046, 10.1029/2001JD000779, 2002.

Fuchs, H., Dubé, W. P., Lerner, B. M., Wagner, N. L., Williams, E. J., and Brown, S. S.: A Sensitive and Versatile Detector for Atmospheric NO2 and NOx Based on Blue Diode Laser Cavity Ring-Down Spectroscopy, Environm. Sci. Technol., 43, 7831-7836, 10.1021/es902067h, 2009.

Mielke, L. H., and Osthoff, H. D.: On quantitative measurements of peroxycarboxylic nitric anhydride mixing ratios by thermal dissociation chemical ionization mass spectrometry, Int. J. Mass Spectrom., 310, 1-9, 10.1016/j.ijms.2011.10.005, 2012.

Phillips, G. J., Pouvesle, N., Thieser, J., Schuster, G., Axinte, R., Fischer, H., Williams, J., Lelieveld, J., and Crowley, J. N.: Peroxyacetyl nitrate (PAN) and peroxyacetic acid (PAA) measurements by iodide chemical ionisation mass spectrometry: first analysis of results in the boreal forest and implications for the measurement of PAN fluxes, Atmos. Chem. Phys., 13, 1129-1139, 10.5194/acp-13-1129-2013, 2013.

Rollins, A. W., Smith, J. D., Wilson, K. R., and Cohen, R. C.: Real Time In Situ Detection of Organic Nitrates in Atmospheric Aerosols, Environm. Sci. Technol., 44, 5540-5545, 10.1021/es100926x, 2010.

Slusher, D. L., Huey, L. G., Tanner, D. J., Flocke, F. M., and Roberts, J. M.: A thermal dissociation-chemical ionization mass spectrometry (TD-CIMS) technique for the simultaneous measurement of peroxyacyl nitrates and dinitrogen pentoxide, J. Geophys. Res., 109, D19315, 10.1029/2004JD004670, 2004.

Sobanski, N., Schuladen, J., Schuster, G., Lelieveld, J., and Crowley, J. N.: A five-channel cavity ring-down spectrometer for the detection of NO2, NO3, N2O5, total peroxy nitrates and total alkyl nitrates, Atmos. Meas. Tech., 9, 5103-5118, 10.5194/amt-9-5103-2016, 2016.

Thieser, J., Schuster, G., Schuladen, J., Phillips, G. J., Reiffs, A., Parchatka, U., Pöhler,

D., Lelieveld, J., and Crowley, J. N.: A two-channel thermal dissociation cavity ring-down spectrometer for the detection of ambient NO2, RO2NO2 and RONO2, Atmos. Meas. Tech., 9, 553-576, 10.5194/amt-9-553-2016, 2016.

Wild, R. J., Edwards, P. M., Dube, W. P., Baumann, K., Edgerton, E. S., Quinn, P. K., Roberts, J. M., Rollins, A. W., Veres, P. R., Warneke, C., Williams, E. J., Yuan, B., and Brown, S. S.: A Measurement of Total Reactive Nitrogen, NOy, together with NO2, NO, and O3 via Cavity Ring-down Spectroscopy, Environm. Sci. Technol., 48, 9609-9615, 10.1021/es501896w, 2014.

Womack, C. C., Neuman, J. A., Veres, P. R., Eilerman, S. J., Brock, C. A., Decker, Z. C. J., Zarzana, K. J., Dube, W. P., Wild, R. J., Wooldridge, P. J., Cohen, R. C., and Brown, S. S.: Evaluation of the accuracy of thermal dissociation CRDS and LIF techniques for atmospheric measurement of reactive nitrogen species, Atmos. Meas. Tech., 10, 1911-1926, 10.5194/amt-10-1911-2017, 2017.

Zheng, W., Flocke, F. M., Tyndall, G. S., Swanson, A., Orlando, J. J., Roberts, J. M., Huey, L. G., and Tanner, D. J.: Characterization of a thermal decomposition chemical ionization mass spectrometer for the measurement of peroxy acyl nitrates (PANs) in the atmosphere, Atmos. Chem. Phys., 11, 6529-6547, 10.5194/acp-11-6529-2011, 2011.

---

## Referee Comment (RC3) · Anonymous Referee #3 · 19 Jul 2020

This manuscript does an overall good job of describing an instrument designed to measure NOx, NOy, and particulate nitrate by thermal dissociation – CRDS. The most useful aspect of the work is the demonstration of problems with the use of activated carbon denuders for removing gas-phase NOy compounds. This will be of great use to many other researchers who use these types of denuders and activated carbon in general!

I recommend it be published after addressing the minor comments below.

[Figure]

The detection limits are listed in the abstract ( 98 ppt for NOx with 1 min averaging) but strangely are not described elsewhere in the manuscript. Is this for a signal-to-noise ratio of 2? 3? How the LOD is defined and these numbers are determined should be in the main text somewhere. Given how sensitive CRDS can be to NO2, I am surprised that the LODs are as high as they are – I would have expected that with a minute of averaging the LOD would be quite a bit lower. Is this a result of the large correction (116 ppt) that must be made to account for the difference in Rayleigh scattering when sampling humid ambient air vs. dry zero air? In addition to that correction that must be made to account for the differences in humidity between sampling and zero measurements, doesn't the change in humidity also change the reflectivity of the mirrors (due to the change in the index of refraction of air), and thus the ring-down times?

The zero air used for zeroes is "CAP 180, Fuhr GmbH"- please clarify what this means – is it compressed zero air from a cylinder, or is it from a zero air generator? Rather than deal with the effects of ambient sampling vs. dry zeroes, why not use humidity-matched air? (e.g., ambient air that has been scrubbed of NO2 via purafil or a catalyst?)

pg 6, last line - define BET pg 13, " However, when the main dilution flow was humidified significant," This sentence appears to missing a word. Or perhaps the last word should actually be "significantly".
* * *

---

## Author Comment (AC1) · 17 Aug 2020

**Author's Response to Referee #1**

In this response, the referee comments (in black) are listed together with our replies (in blue) and the changes to the original manuscript (in red).

This paper reports the development of an instrument to measure NO2, total NOy and total particulate nitrate based on Cavity Ring-Down spectroscopy for measuring NO2 and thermal dissociation for NOy. The accurate measurement of NOx down to low (ppt) levels is crucial for understanding the chemistry of remote atmosphere and combining such an instrument with thermal dissociation to measure total reactive nitrogen compounds and particulate nitrate further adds to the potential uses of such an instrument. Whilst a few examples of this type of approach exist in the literature, CRDS is a relatively new method and so work like this is important.

In general the paper details a comprehensive laboratory study of the instrument, including the thermal decomposition of different NOy species and the performance of the denuder for the particulate measurements. It is well written, easy to follow and within scope of the journal. I recommend publication subject to the following, largely minor amendments and additions.

We thank the referee for the positive review of our paper and the constructive comments, which we address in the following responses.

A detailed description of the CRDS NO2 / NOx instrument is given in a previous paper (Thieser et al 2016), however I feel this new paper would benefit from some more details on the performance of the instrument to NO2 and NO. There is no mention of how these species are calibrated, or what the precision / accuracy are. Whilst this data may be able to be found elsewhere, I believe the authors should include it here as well. It would greatly assist readers wishing to get a full understanding of the performance of the instrument. I would suggest at least adding what calibration gases were used for NO and NO2, what is the accuracy and precision of these measurements at various time resolutions and what is the magnitude of any interferences.

We added the following paragraph to Sect. 2.2 as a short overview of the instrument's performance in detecting  $NO_2$  and NO.

For NO2, the performance of the instrument was first described by Thieser et al. (2016), who reports a measurement uncertainty of 6 % + (20 pptv\*RH/100) which is dominated by uncertainty in the effective cross section of NO2 and the wavelength stability of the laser diode. The NOx detection limit of 40 pptv ( $2\sigma$ , 1 minute average) for the present instrument (laboratory conditions) was derived from an Allan variance analysis and is worse than that reported by Thieser et al. (2016) (6 pptv at 40 s) due to degradation of the mirror reflectivity. Corrections applied to take into account humidity and pressure changes are discussed in Sect. 2.1. The total uncertainty in NOy will depend on the uncertainty in the conversion to NOx of both gaseous and particulate nitrate and thus depends on the individual components of NOy in the air sampled. For purely gaseous NOy, the major problem is likely to be related to loss of sticky molecules at the inlet and we choose to quote a "worst case" uncertainty of 15%.

We have amended the LOD we quote to that obtained on a stationary platform (the one mentioned in the last version was derived from the AQABA dataset obtained on a ship):

In this context we note that the deployment on a ship resulted in a degradation in performance (LOD was  $\approx$  100 pptv) owing to the ship's motions, especially in heavy seas, which resulted in drifts in the instrument zero.

On page 8 lines 17-20 it is stated that complete conversion of HNO3 to NO2 occurs at temperatures above 800°C, but then that the amount of NO2 detected of 13 ppb is 85% of that expected based on the permeation and dilution flows. These two things do not seem to be consistent with each other – could the authors please clarify? Also, no mention is made of any potential losses of HNO3 to the surface of the instrument or the inlet, something that is often a problem with this type of instrument?

The uncertainties of the measurements have to be taken into account. We have modified our text and now write:

A custom-made permeation source was used to provide a constant, known flow of  $HNO_3$  (with ~ 8%  $NO_X$  impurity) to the TD-CRDS inlet. The permeation source consisted of a length ( $\approx$  1m) of PFA tubing immersed in 66% HNO3 solution held at 50 °C through which 100 sccm of dry, zero-air was passed. The concentration of HNO3 and thus its permeation rate,  $(1.62 \pm 0.2) \times 10^{-4}$  sccm, was derived by measuring the optical extinction of HNO3 at 185 nm using the absorption cross section of Dulitz et al. (2018). The uncertainty is related to uncertainty in the absorption cross-section and the reproducibility of the output. The HNO3 thermogram (Fig. 2 and Fig. S2c)) has a plateau at temperatures above  $\approx$  800 °C. In the plateau region of Fig. 2, the HNO3 mixing ratio measured is  $13.0 \pm 0.8$  ppb, which (within combined uncertainties) is in agreement with the expected value (15.2  $\pm$  1.98 ppb) calculated from the permeation rate and uncertainty in the dilution factor. We cannot rule out some loss of HNO3 in the tubing connecting the permeation source to the TD-CRD, through previous studies have shown that irreversible losses are ~ of 5% or less under dry conditions (Neuman et al., 1999). We note that inlet loss of HNO3 is minimized under ambient sampling conditions as only a short section (~20 cm) quartz tubing at ambient temperature is upstream of the heated section in which HNO3 is converted to NO2. Our observations are thus in accord with previous studies that found complete conversion of HNO3 to NO2 in similar set-ups (Day et al., 2002; Di Carlo et al., 2013; Wild et al., 2014; Womack et al., 2017).

Could the authors comment on if there would be an effect of HONO on the NOy channel?

We have performed additional experiments with other trace gases, including HONO. Our results indicate efficient conversion of HONO to  $NO_x$  in our heated inlet and are described in a new section 3.1.5.

In section 3.3.1 could the authors make some comment as to how much particles greater than 414 nm in diameter are transmitted? I would have thought that, especially in remote marine environments, particulate nitrate have a significant fraction on larger particles and thus provide an interference to the instrument.

We have addressed this comment by adding text in Sect. 3.3.1:

The PLC does a better job in predicting a reduction in transmission for the largest particles which we measured and indicates a transmission of 74% at 1  $\mu$ m and 45 % at 2  $\mu$ m. In certain environments, nitrate associated with coarse mode particles thus represents a potential bias for TD-CRDS measurements of NOy.

In section 3.3.2 could the authors comment on how the efficiency of the denuder changes with age and how often it may need to be regenerate or replaced.

This will depend on the conditions of its deployment (e.g. highly polluted or remote) and we cannot suggest a regeneration schedule. Also, the behavior of our denuder is not necessarily identical to that of other designs.

In section 4.1 the authors state that they do not present an analysis of the NOz data as it will be presented in a future publication. I think they should at least comment on the NOz data observed. This paper is about the development of an instrument to measure NOz and to not comment on the measurements made even a little seems very strange.

Unlike  $NO_x$ , there were no measurements during AQABA with which to directly compare our  $NO_z$  data set. The separate publication will be non-technical and will deal with the atmospheric chemistry of  $NO_z$  which is not in the scope of AMT. We see little value in reproducing information from the planned publication here. We now write:

Here we compare the  $NO_x$  and pNit measurements with other measurements of these parameters made during the campaign.

Likewise section 4.2 would also benefit from an expanded discussion of the NOz data. For instance why is the diurnal cycle observed as it is, especially the nighttime peak values.

We expanded the discussion and now write:

The diel profiles of NOz/NOy are strongly influenced by fresh emissions of NOx. As the measurement location is strongly influenced by traffic, there is a decrease in NOx (and increase in NOz/NOy) at nighttime. Nightime increases in NOz ( $13^{th}-14^{th}$ ,  $15^{th}-16^{th}$ ,  $18^{th}-19^{th}$  and  $19^{th}-20^{th}$  of January 2020) may also be partially caused by formation of N2O5 as previously observed (Schuster et al, 2009) and which would have been favoured by the low nighttime temperatures (< 10 °C) in winter.

Finally I wonder if the authors could comment on now the particulate nitrate measurement could be improved. There are some suggestions given in section 3 but I think there should be something in the conclusions about this. Currently I read the paper like there was not much hope that the technique could be used for accurate particulate nitrate measurements but I am sure this is not the case, thus the authors should say so.

The following text has been added to the conclusions:

Under humid conditions the denuder suffered from direct breakthrough of NO and the re-release of previously stored iPN and  $NO_2$  in the form of NO, indicating a potential bias of pNit measurements using this technique and potentially limiting its deployment to low- $NO_x$  and low- $NO_z$  environments. When using comparable denuders, we recommend regular checks with humidified zero air to characterize potential breakthrough. Our experiments demonstrated that the release of  $NO_x$  from the denuder exposed humid zero-air for several hours can decrease to values below 1 ppbv, which, in a first approximation could be treated as an offset. Cycling between multiple denuders would help in reducing the size of any bias.

Page 13 line 6: 'humidified significant' does not make sense.

Typo has been removed humidified significant

**References**

Day, D. A., Wooldridge, P. J., Dillon, M. B., Thornton, J. A., and Cohen, R. C.: A thermal dissociation laser-induced fluorescence instrument for in situ detection of NO2, peroxy nitrates, alkyl nitrates, and HNO3, J. Geophys. Res. -Atmos., 107, doi:10.1029/2001jd000779, 2002.

Di Carlo, P., Aruffo, E., Busilacchio, M., Giammaria, F., Dari-Salisburgo, C., Biancofiore, F., Visconti, G., Lee, J., Moller, S., Reeves, C. E., Bauguitte, S., Forster, G., Jones, R. L., and Ouyang, B.: Aircraft based four-channel thermal dissociation laser induced fluorescence instrument for simultaneous measurements of NO2, total peroxy nitrate, total alkyl nitrate, and HNO3, Atmospheric Measurement Techniques, 6, 971-980, 10.5194/amt-6-971-2013, 2013.

Dulitz, K., Amedro, D., Dillon, T. J., Pozzer, A., and Crowley, J. N.: Temperature-(208-318 K) and pressure-(18-696 Torr) dependent rate coefficients for the reaction between OH and  $HNO_3$ , Atmos. Chem. Phys., 18, 2381-2394, 2018.

Neuman, J. A., Huey, L. G., Ryerson, T. B., and Fahey, D. W.: Study of inlet materials for sampling atmospheric nitric acid, Env. Sci. Tech., 33, 1133-1136, 1999.

Wild, R. J., Edwards, P. M., Dube, W. P., Baumann, K., Edgerton, E. S., Quinn, P. K., Roberts, J. M., Rollins, A. W., Veres, P. R., Warneke, C., Williams, E. J., Yuan, B., and Brown, S. S.: A measurement of total reactive nitrogen, NOy, together with  $NO_2$ , NO, and  $O_3$  via cavity ring-down spectroscopy, Environmental Science & Technology, 48, 9609-9615, doi:10.1021/es501896w, 2014.

Womack, C. C., Neuman, J. A., Veres, P. R., Eilerman, S. J., Brock, C. A., Decker, Z. C. J., Zarzana, K. J., Dube, W. P., Wild, R. J., Wooldridge, P. J., Cohen, R. C., and Brown, S. S.: Evaluation of the accuracy of thermal dissociation CRDS and LIF techniques for atmospheric measurement of reactive nitrogen species, Atmospheric Measurement Techniques, 10, 1911-1926, 10.5194/amt-10-1911-2017, 2017.

---

## Author Comment (AC2) · 17 Aug 2020

**Author's Response to Referee #2**

*In this response, the referee comments (in black) are listed together with our replies (in blue) and the changes to the original manuscript (in red).*

Friedrich et al. describe a 2-channel thermal dissociation cavity ring-down spectrometer (TD-CRDS) for quantification of NOx, NOy, and particulate nitrate (pNit). Thermograms of peroxyacyl nitrate (PAN), isopropyl nitrate and nitric acid are presented. The potential interference from NH3 and secondary radical chemistry is evaluated. The use of an activated carbon denuder to suppress gas-phase components of NOy and transmit pNit is described. Several experiments are presented to characterize the performance of this denuder: the transmission of ammonium nitrate particles in the 10 nm - 414 nm size range and the partial removal of several trace gases (NO, NO2, PAN, iPN, and HNO3) as a function of relative humidity (RH) which revealed inlet memory effects. Sample ambient air measurements from the 2017 AQABA campaign and ambient air measurements in Mainz, Germany are presented.

This research group has described similar instruments previously (Sobanski et al.,2016; Thieser et al., 2016) and in this paper extends the earlier measurement capabilities to now quantify NOx, NOy, and aerosol nitrate. The measurement of NOx and NOy (including NH4NO3) has previously been demonstrated by Fuchs et al. (2009) and Wild et al. (2014), so that the main novelty of this work is the use of the denuder to selectively quantify ammonium nitrate. A denuder was previously used by the Cohen group (Rollins et al., 2010) to quantify organic nitrates which dissociate at a lower inlet temperature than NH4NO3.

The measurement of NOx by CRDS through the addition of O3 is convincing. However, the paper will require considerable revision before it can be considered for acceptance. I have summarized my major concerns below. Most importantly, I am not convinced that the instrument presented here yields accurate NOy and particulate nitrate data.

We thank the referee for the comprehensive and constructive comments on our paper, which we address in the following responses. We have performed additional laboratory experiments, as suggested.

Major comments

(1) Thermograms and temperature dependence of thermal dissociation:

(a) The thermograms presented in this paper are inconsistent with literature, but no convincing rationale is provided as to why that would be. (Day et al., 2002; Wild et al., 2014) have shown that thermograms of iPN and HNO3 are considerably offset from each other, which is what one would expect from their Arrhenius parameters (and is shown in the SI of this paper as Figure S2D). The "rather short" heated section of the inlet likely broadens the thermograms which is undesirable but should not have resulted in their complete overlap.

We re-measured the iPN thermogram using a thoroughly cleaned canister and with a different liquid sample. The newly measured thermogram is much closer to that expected from our own previous measurements and we conclude that the previous sample was contaminated (presumably with $HNO_3$ as the referee suggested). Note that while this result is reassuring, it changes none of the conclusions drawn in the manuscript as the instrument is not designed to measure PNs, ANs, $HNO_3$ etc separately, but to provide a measurement of $NO_y$. We now write:

A 10 L stainless-steel canister containing 10.3 ppmv of isopropyl nitrate (iPN) at a pressure of 4 bar $N_2$ was prepared using a freshly vacuum-distilled liquid sample using standard manometric methods. $NO_x$ impurities were ~ 4.7 ppbv, though we note that diluted iPN stored in stainless-steel canisters for periods of several weeks degrades to form $NO_2$ and $HNO_3$.

The thermogram is displayed in Fig. 2, the absolute concentrations in Fig. S2b). Based on the mixing ratio of iPN in the canister and the dilution flows, 10.7 ppbv represents (101 $\pm$ 11)% conversion. The shaded area around the expected iPN mixing ratio in Fig. S2b) signifies the uncertainty of this value, based on propagation of the errors during the manometric and dilution procedures (2% for flow rates, 5% for pressures measured with digital pressure gauges and 10% for the last dilution step using the analog pressure gauge of the canister).

Between 550 and 850 °C we observe a weak increase in $NO_2$ from 10.7 to 11.2 ppbv, which is likely due to small amounts of $HNO_3$ in the sample. For iPN, the temperature at 50% conversion is 50 °C higher than those reported by Thieser et al. (2016) and Sobanski et al. (2016). Wild et al. (2014) employed a gaseous mixture of different alkyl nitrates and also observed a broad thermogram, with a an initial increase in $NO_2$ (up to 80% conversion) for temperatures < 300 °C, followed by a slower increase up to 800 °C. The alkyl nitrates thermogram of Wild et al. (2014) has been included into Fig. S2b) to illustrate this behaviour and to facilitate direct comparison.

 (b) The thermogram for PAN is inconsistent with a large body of literature including work by the Mainz group (Phillips et al., 2013) which showed full dissociation of PAN at an inlet temperature of ~150 °C (not at ~400 °C). If the higher dissociation temperatures are a consequence of the short heater residence time, then this would be design flaw as the higher temperatures enable unwanted side reactions, increase thermal gradients, and reduce selectivity.

This instrument is not designed to be selective and separately measure PAN and ANs. It is designed to measure $NO_y$ and the positions of the individual thermograms are not of central importance as long as the plateau region is reached. The Phillips 2013 instrument used a PFA tube wrapped with heating wire rather than a quartz tube inserted into a commercial oven and was operated at a lower flow rate. There is no reason to expect that thermograms obtained at different flows with different tube materials and diameters (and thus temperature gradients) should be similar. In order to more fully understand the source of the differences which the referee has highlighted, we have measured the gas temperature by inserting a thermocouple into the oven region. We found, for example that with the oven temperature set at 310 °C, the thermocouple reading was just 230 °C, which helps explain the shift to higher temperatures when using the present set-up. We have also assessed the impact of

thermal gradients on the thermograms in another supplementary figure (see Fig. S3c)). The following text has been added:

By inserting a thermocouple into the middle part of the heated section under normal sampling conditions we were able to show that the temperature of the gas was $\approx$ 80 °C lower than that indicated by the oven's internal temperature sensor in the 200-300 °C temperature range and about 40 °C lower at a set temperature of 600 °C (see Fig. S3a). We were unable to measure the temperatures of the gas stream at oven temperatures above about 600 °C and throughout the manuscript we refer only to the temperature indicated by the internal sensor of the oven.

We have added a new general section explaining why thermograms measured in different setups may differ.

**3.1.8 Summary of thermograms**

The thermograms obtained by the present instrument deviate from others reported in the literature, the temperatures required for 50% dissociation being generally higher by e.g. 80 °C for PAN, 50 °C for iPN and 150 °C for $HNO_3$, respectively (Day et al., 2002; Wild et al., 2014; Sobanski et al., 2016; Thieser et al., 2016; Womack et al., 2017). This lack of agreement with other setups is not unexpected as the degree of dissociation of a trace gas at any temperature depends not only on the temperature but also on the time over which the molecule is exposed to that temperature (Womack et al., 2017). To illustrate this, based on rate coefficients (related to bond-dissociation energies, BDE) for the thermal dissociation of PAN (Bridier et al., 1991), iPN (Barker et al., 1977), $HNO_3$ (Glänzer and Troe, 1974), $N_2O_5$ (IUPAC, 2019), $ClNO_2$ (Baulch et al., 1981), and HONO (Tsang and Herron, 1991), we calculated the theoretical 50% conversion temperature for each molecule as a function of residence time inside the oven (see Fig. S3b)). At short residence times the dependence on temperature is very steep (especially for large BDEs) which partially explains the differences between our short heated section inlet and longer ones. However, in practise, we know neither the precise average temperature of the gas at the centre of the oven, nor can we characterise the axial and radial gradients in temperature in the quartz tubes so that calculations of fractional dissociation (or complete thermograms) based on bond-dissociation energies are at best only a rough guide. We note that use of different flows, oven diameters and operational pressures will strongly affect heat transfer from the oven walls to the gas, so that reporting the temperature of the external oven-wall (as done here and in all reports in the literature) to some extent precludes comparison between different setups. The width of the thermograms (i.e. the temperature difference between e.g. 10% and 90% dissociation) will also depend on details of axial and radial temperature gradients in the tubing located within the oven and also in the downstream section of tubing, which represents a transition regime between oven and room temperature. The impact of temperature gradients inside the quartz tube was explored by calculating the $HNO_3$ thermogram using an Arrhenius expression for its thermal dissociation and the gas residence time within the quartz tube. First we assumed that all $HNO_3$ molecules experience the same temperature and then compared this to the situation in which 20% of the $HNO_3$ molecules are 80 °C lower in, and 20% are 80 °C higher in temperatures. The resultant thermograms are displayed in Fig S3c) and indicate that the presence of temperature gradients results in an increase in the width of the thermogram from 250 °C to 350 °C.

The thermograms we report here serve only to determine the temperature needed to ensure  maximum conversion of each trace gas to $NO_2$. This is achieved in the present setup with a temperature of 850 °C. Where possible, we have verified that operation at the plateau of the thermogram resulted in quantitative conversion of the traces gases and particles studied, with one exception, $NaNO_3$ particles. We further note that, in an instrument designed only to measure $NO_y$, there is no need to ensure separation (in temperature) of the thermograms for different classes of molecules.

(c) Have the authors independently confirmed the identity, purity and concentrations of the gases they sampled by CIMS (PAN, HNO3) or GC (iPN)? For PAN, Figure S2a shows a step after ~300 °C which may be due to the presence of an alkyl nitrate. For iPN and HNO3 Figure S2 shows "expected values" based on permeation and dilution flows, though I consider these methods reliable standards.

As the referee mentions, flows of $HNO_3$ and iPN were obtained from either permeation standards or canisters with known mixing ratios. Experiments on $N_2O_5$, carried out following a suggestion of this referee (see comment below), were conducted with parallel sampling by a further TD-CRD operated in this laboratory. Experiments on PAN, $ClNO_2$ and HONO were conducted without parallel measurements by other instruments. Note that TD methods have been used in the past to calibrate e.g. PAN and $ClNO_2$ signals for a CIMS, so reversing the logic and using a CIMS to calibrate the TD instrument in this work makes little sense.

We would like to re-emphasise that our instrument is not set-up to measure different members of the $NO_z$ family but to measure $NO_y$. An impurity e.g. of $HNO_3$ in the iPN sample (see above) will surely change the shape of the thermogram, but will not prevent conversion of all $NO_z$ to $NO_2$ and NO at the operational temperature of 850 °C. Text (section 3.1.8) has been added to emphasise this:

The thermograms we report here serve only to determine the temperature needed to ensure  maximum conversion of each trace gas to $NO_2$. This is achieved in the present setup with a temperature of 850 °C. Where possible, we have verified that operation at the plateau of the thermogram resulted in quantitative conversion of the traces gases and particles studied, with one exception, $NaNO_3$ particles. We further note that, in an instrument designed only to measure $NO_y$, there is no need to ensure separation (in temperature) of the thermograms for different classes of molecules.

(d) The thermogram of NH4NO3 should be shown.
(e) N2O5, ClNO2, and HONO are important NOy components; their thermal dissociation should have been evaluated since the claim is made that the inlet's behaviour is different from that of Wild et al. (2014).

We have performed additional laboratory experiments, and now include thermograms for $NH_4NO_3$, $N_2O_5$, $ClNO_2$ and HONO.
See new Sections 3.1.4 to 3.1.7.

(2) Denuder performance:

(a) The denuder partially transmits gases but at a rate that appears to be dependent on environmental factors such as RH (Figure 6) and possible also ambient air temperature (not examined). The transmission of many potentially important components of NOz (e.g., N2O5, ClNO2, HONO) was not evaluated but should have been.

We have performed additional laboratory experiments with the denuder, and now include $N_2O_5$, $ClNO_2$ and HONO in Fig. 6. The removal efficiency of $N_2O_5$ was only determined in dry air, to prevent the formation of $HNO_3$. $ClNO_2$ and HONO were only tested under humid conditions, as humidity was required for their initial generation. The text has been modified:

The efficiency of removal of trace gases in the denuder under typical flow conditions (3.3 slm) was investigated for NO, $NO_2$, PAN, iPN, HONO, $N_2O_5$, $ClNO_2$ and $HNO_3$ as representative $NO_y$ species. The efficiency of removal of each trace gas (generally present at 5-40 ppbv) was determined by measuring its relative concentration when flowing through the denuder (pNit-channel) and when bypassing the denuder ($NO_y$ channel). The results (Fig. 6) indicate that, in dry air, all of these trace gases were removed with an efficiency of close to 100%. However, when the main dilution flow was humidified significant, RH-dependent breakthrough of NO was observed, with only 60% stripped from the gas-phase at RH close to 100%. HONO was removed with 85% efficiency at an RH of 46%, and $ClNO_2$ with 75% efficiency at an RH of 60%. In contrast, humidification had only a marginal effect on the scrubbing efficiency for $NO_2$, iPN and $HNO_3$ for which an efficiency of $\geq$ 95% was observed. The precise values from which the removal efficiencies in Fig. 6 were determined are listed in Table S2.

(b) Memory effects. The high-concentration experiments described on pg 13 are a poor way of assessing memory effects because there could be a limited number of surface sites that are overloaded when sampling a high concentration. A more realistic experiment would be to sample a low concentration of NOx for a longer time period, and then examine what comes off the denuder while sampling zero air.

We do not aim to provide a quantitative analysis of memory effects but to indicate that such effects can be important for this denuder type. We also have data in which the denuder was exposed to "typical" concentrations of $NO_x$ for long periods but under less controlled (i.e. more variable conditions). The qualitative result, a release of $NO_x$ when exposed to humidity, is the same. We now write:

Qualitatively similar results, i.e. humidity induced formation and release of $NO_X$ from the denuder, were observed when the denuder was exposed for periods of weeks to variable levels of $NO_x$ (i.e. up to 20 ppbv) under dry conditions.

(c) NOx and the aforementioned components of NOz usually dominate NOy, such that any transmission of these species constitutes a sizeable measurement error for pNit. This should have been taken into account in an explicit error analysis as part of the pNit data reduction.

After identifying the humidity issues of the denuder and in the absence of effective regeneration techniques, we refrained from detailed analysis (including assessment of errors) in our pNit measurements. Our findings, that the denuder suffers from humidity related breakthrough and release of reactive nitrogen indicates that measurements of pNit using similar denuders to remove gaseous $NO_y$ may suffer from bias under some conditions. We are unable to quantify this. In the ambient measurements from Mainz we measured only total $NO_y$ and did not attempt to separate between gas and particulate phase $NO_y$. The pNit measurements from AQABA serve to demonstrate the presence of humidity issues under campaign conditions and were not included as part of an analysis of gas-particle nitrogen partitioning. We write:

This strongly suggests that the large difference between pNit reported by the TD-CRDS and the AMS does not results from the inability of the AMS to detect supermicron particulate nitrate, but from denuder artefacts similar to those seen in the laboratory experiments described in Sect. 3.3.2. This short case-study serves to highlight the

potential positive bias in denuder based, TD-CRDS measurements of pNit under humid field conditions.

(3) Aerosol nitrate can be present on mineral dust as organic nitrates or on sea salt aerosol. Do other nitrate salts convert to NO2 in this inlet?
We have performed extra experiments on $NaNO_3$ and compared the detection efficiency of $NaNO_3$ with that of $NH_4NO_3$ as a function of particle diameter.
The results are presented and discussed in the new section 3.1.7

(4) How does the instrument perform above the 1-micron size range? The latter dominates aerosol mass in many regions.
As we did not carry out experiments with super-micron particles, we can only assess this using the predicted transmission of the denuder according to the PLC. The following paragraph has been added in Sect. 3.3.1:
The PLC does a better job in predicting a reduction in transmission for the largest particles which we measured and indicates a transmission of 74% at 1 µm and 45% at 2 µm. In certain environments, nitrate associated with coarse mode particles represents a potential (negative) bias to TD-CRDS measurements of $NO_y$.

(5) Ambient air measurements of pNit:
(a) How was the error due to break-through of NOx considered?
(b) How were denuder memory effects taken into account in the reduction of the field data?
(c) The RH dependence of denuder performance is a considerable issue considering the inlet is periodically flooded with dry zero air and then re-exposed to humid ambient air. How does this back-and-forth affect the field data?
See response to major comment (2) (c). We do not show pNit data sets that were corrected for the break-through and that can be considered as accurate pNit measurements. The denuder is never back-flooded with (hot) zero air, as this would melt the PFA fittings between denuder and quartz tube.

(d) How long was the inlet for the NOy channel? Was conductive tubing used (or were aerosol lost on the inlet)?
During AQABA, sampling was directly through the denuder in the pNit channel and through the heated quartz inlet in the $NO_y$ channel, so that aerosol losses to non-conductive tubing or bent inlet lines was avoided. In the Mainz ambient measurements, the pNit channel was not operated and the $NO_x$ and $NO_y$ channels sampled from a common overflow in a straight ca. 1 m long ½ inch PFA tube. Aerosol loss in the $NO_y$ channel might be possible under this setup. We added in Sect. 4.2:
Aerosol transmission was probably < 100% in these measurements.

(e) Since the TD-CRDS data were not compared with an independent measurement of NOy (Mo converter CL) or of total aerosol nitrate (High-volume impactors, PILS, MARGA or similar), the claim that this instrument accurately quantifies NOy and particulate nitrate in ambient air is not substantiated and the conclusions need to be weakened accordingly.
In the conclusions we now write:
Our laboratory experiments suggest that the different gas-phase $NO_z$ species investigated (PAN, iPN, $N_2O_5$, HONO, $ClNO_2$, $HNO_3$) are converted with near stoichiometric efficiency to $NO_x$ at an oven temperature of 850 °C. $NH_4NO_3$ particles of diameter 200 nm are also detected quantitatively as $NO_x$, whereas the efficiency of

detection of NaNO$_3$ particles of similar diameter was closer to 25%. The efficiency of detection of coarse mode particles will be further reduced by their lower transmission through the denuder.

(6) The authors show several pages on the interference due to gas-phase NH3 only to conclude it to be insignificant. However, much more NH3 is typically present in the form of aerosol ammonium, which would evaporate in the NOy inlet. The authors should also consider and examine conversion of (NH4)2SO4 especially in the presence of O3. The concentration of NH$_3$ which we used (131 ppb) corresponds to an aerosol ammonium loading of about 100 microg m$^{-3}$, which is more than found in most environments. As there is no obvious reason why NH$_3$ from (NH$_4$)$_2$SO$_4$ should behave differently to gas phase NH$_3$. We do not see any value in performing experiments on (NH$_4$)$_2$SO$_4$.

(7) General organization:
(a) The introduction with its lengthy discussion of nitrogen oxide chemistry misses the mark (see detailed comment below).
We would argue that some description of NO$_y$ chemistry is essential to put this work in context and describe our motivation for the development of the instrument. Too much is probably better than too little and the well informed reader has the choice of simply skipping this section. See specific comment below for changes made.

(b) A critical comparison of the performance of this new instrument to existing methods (in terms of detection limits, selectivity, instrument including heater designs, etc.) is lacking and should be provided.
We have added an additional section (3.4) in which we compare the instrument to others. Note that we restrict this discussion to instruments that measure NO$_x$ and NO$_y$ and not single components thereof.

Specific comments
pg 1 line 10 "detection" replace with quantification
"detection" has been replaced with "measurement"

pg 1 line 14 "detection limits" Please state the level of confidence. Why does the NOy channel have half of the LOD of the NOx channel?
Our detection limits were based on instrument performance during the AQABA campaign in which the ship's motion degraded optical alignment and resulted in great variability in the zeros. We now report the LOD of both channels during operation in the laboratory, which is comparable to any stable platform.
*Abstract:* Detection limits, defined as the 2$\sigma$ precision for 1 minute averaging, are 40 pptv for both NO$_x$ and NO$_y$.
*Sect. 2.2:* The NO$_x$ detection limit of 40 pptv (2$\sigma$, 1 minute average) for the present instrument (laboratory conditions) was derived from an Allan variance analysis and is worse than that reported by Thieser et al. (2016) (6 pptv at 40 s) due to degradation of the mirror reflectivity.
*Sect. 3.4:* Our present detection limit for both NO$_x$ and NO$_y$ is however worse than that reported (for NO$_2$) for the same instrument in 2016 (Thieser et al., 2016), which is a result of mirror degradation since that study.

pg 1 line 17, 18, 19 "significant interferences", "high particle transmission", "essentially complete removal" Please be quantitative.

Corrected

[…] and rule out significant interferences from $NH_3$ detection (< 2%) or radical recombination reactions under ambient conditions. While fulfilling the requirement of high particle transmission (> 80% between 30 and 400 nm) and essentially complete removal of reactive nitrogen under dry conditions (> 99%), […]

pg 1 line 19 "denuder suffered from NOx breakthrough" Does the breakthrough of NOx not imply that the pNit measurement is inaccurate and does not work?

The breakthrough certainly indicates a potential bias. See replies to major comments (2) (c) and (5) (c). In the abstract we write:

The denuder suffered from $NO_x$ breakthrough and memory effects (i.e. release of stored $NO_y$) under humid conditions, which may potentially bias measurements of particle nitrate.

pg 1 line 21-22 "NOx measurements obtained from a ship sailing through the Red Sea, Indian Ocean and Arabian Gulf .... were in excellent agreement with those taken by a chemiluminescence detector of NO and NO2." What about the NOy and pNit data during this cruise?

The oven of the pNit channel broke down shortly after the time frame shown in Sect. 4.1. An analysis of the $NO_y$ data set will be presented in a separate publication.

pg 1 line 23 "A dataset exploring variations in the NOz to NOy ratio (maximum value of 0.6) of air in a region (Mainz, Germany) with strong urban influence was measured over a one-week period in winter." and what was the conclusion?

We now write:

Summertime $NO_x$ measurements obtained from a ship sailing through the Red Sea, Indian Ocean and Arabian Gulf ($NO_x$ levels from < 20 pptv to 25 ppbv) were in excellent agreement with those taken by a chemiluminescence detector of NO and $NO_2$. A dataset obtained locally under vastly different conditions (urban location in winter) revealed large diel variations in the $NO_z$ to $NO_y$ ratio which could be attributed to the impact of local emissions by road-traffic.

pg 2 line 1 - pg 3 line 21 Section 1.1 "Atmospheric NOx and NOy". This paper is about a new instrument. The lengthy description of NOx and NOy chemistry and all those reactions (in particular R6-R10) are not needed and could / should be replaced by citations to the authors' own papers.
If this section stays, please fix R8. RH is already used as relative humidity and implies a saturated alkane, with which NO3 barely reacts and usually does not form an alkyl nitrate.

See reply to major comment (7) (a). We replaced "RH" with a generic alkene.

pg 4 line 17 "Pyrolysis". Thermolysis or thermal dissociation are more appropriate here.
Changed throughout the manuscript.

pg 4 line 25. "which overcome these limitations" Please be more specific here. One of the limitations discussed in the preceding paragraph mentions secondary chemistry by O-atoms formed from the decomposition of ozone, which wasn't addressed in this paper.
More detail has been added:
Compared to the setups described by Thieser et al. (2016) the following changes were implemented: (1) Addition of $O_3$ for $NO_x$ detection; (2) higher oven temperature (to

detect HNO$_3$) and location directly at the front of the inlet; and (3) use of a charcoal denuder for separate measurement of pNit and gas-phase NO$_z$. The addition of O$_3$ (after the TD-inlet) ensures that we detect NO as well as NO$_2$ and thus removes bias caused e.g. by the pyrolysis of O$_3$ and reactions of O($^3$P) which reduce NO$_2$ to NO.

Since some of the co-authors have described multi-channel TD-CRDS instruments previously (Sobanski et al., 2016; Thieser et al., 2016), please add a short statement explaining how this new instrument differs from the old ones and what parts constitutes novelty (use of a denuder to quantify pNit).
Text has been added:
Compared to the setups described by Thieser et al. (2016) the following changes were implemented: (1) Addition of O$_3$ for NO$_x$ detection; (2) higher oven temperature (to detect HNO$_3$) and location directly at the front of the inlet; and (3) use of a charcoal denuder for separate measurement of pNit and gas-phase NO$_z$.

pg 5 Section 2.1 "CRDS Operation Principals" How much of this section is duplication of (Sobanski et al., 2016; Thieser et al., 2016)? Please condense and focus on what has been changed since the earlier versions, and why.
Apart from the core optical NO$_2$ detection most key parameters were changed compared to the setup described by Thieser et al. which did not measure NO$_Y$.

pg 5 line 5 spell out / define STD
Corrected.

pg 5 line 21 Please give an uncertainty for l/d.
Corrected.
$l / d = 0.98 \pm 0.01$

pg 5 line 23 "aerosol particles" should simply be particles.
Corrected.

Does a 2 µm pore size filter truly removes all particles? In our experience, they do not, but they remove the size range that would interfere optically. Consider rephrasing.
We performed a quick check with laboratory air and a CPC and added the results:
The filter's efficiency, tested with laboratory air containing $1.8 \times 10^3$ particles cm$^{-3}$ and a CPC (TSI 3025 A), was > 98%.

pg 5 line 27 "depending on flow, pressure and inlet set-up" State typical pressures and flows. Is the inlet described anywhere? Please call out the relevant section if it is.
Text added.
(see sections 2.2 and 2.3).

pg 5 line 29 300 pptv NO2 equivalent and 6.5 Torr pressure difference seem like a lot. It seems to be caused by the peculiar addition point of the zero air between the valve/denuder/inlet converter and CRDS cells and probably could be avoided altogether if zero air were added at the tip of the inlet (with larger fittings). Please provide a rationale why zero air was added this way (state advantages and disadvantages).
The current set-up was implemented for a chamber study, where flowing hot air into the chamber was undesired. The pressure effect on NO$_2$ detection is well characterised and the correction worked consistently in all field and laboratory situations. We have

also zeroed by overflowing zero-air through the heated inlet (without denuder) and this option is now mentioned.

We have also used an alternative setup, in which the inlet is overflowed with zero air added close to the tip of the inlet (downstream if the oven) reduces the pressure difference, but has the disadvantage that hot air is blown out of the instrument when zeroing, which may interfere with co-located inlets. Addition of zero air upstream of the quartz inlets would remove this problem but increase the complexity of the inlet and potentially result in loss of sticky molecules such as $HNO_3$.

pg 6 line 9. "The maximum concentration of NO2 (and thus optimal conversion of NO to NO2)". Please state what fraction of NO that is converted and if the NO data were corrected accordingly.

We have added the conversion factor. No correction was performed:

The maximum concentration of $NO_2$ (corresponding to 96% of the NO in the gas bottle) was observed when the flow over the pen-ray lamp was between 60 and 80 sccm, which resulted in 19 ppmv $O_3$ in the reaction volumes.

pg 6 Section 2.3 How is temperature measured in these furnaces? Is the oven temperature identical to the temperature of the gas travelling through it?

The temperature is measured in the ceramic block that accommodates the quartz tube. See also the reply to major comment (1) (b) and the new Figure S3a).

sections 2.3 and 2.4. A critical parameter is the sample flow rate, which should be stated here.

Has been added:

The sampling flow through both heated inlets is 3.0 slm.

pg 6 line 20. How short is "short"?
An experiment of the inlet setups during the two field campaigns should be provided.

On AQABA sampling occurred directly through the denuder in the pNit channel and through the heated quartz inlet in the $NO_y$ channel. During the ambient measurements in Mainz, the $NO_x$ and $NO_y$ channels sampled from a common overflow in a straight, 1 m long ½ inch OD PFA tube (as described in Sect. 4.1 and 4.2). Inlet lines, filter location and the pressure reduction were identical in both campaigns.

[…] was kept short (ca. 30 cm) […]

pg 6 line 22. Describe the valve (make & size etc.). Are there memory effects? Pressure drops?

A description has been added:

An electronic, PTFE 3-way valve (*Neptune Research, Inc.*, type 648T032, orifice diameter 4 mm) under software control switches between the two heated inlets, one of which is equipped with a denuder. Memory effects for $NO_2$ on the valve surfaces were not observed. Bypassing the valve under normal sampling conditions led to an 0.6 Torr pressure change.

pg 7 line 11 "Results and discussion, 3.1 The fractional conversion of NOz to NO2 in the TD-inlets was investigated in a series of experiments in which constant flows of (separately) PAN, isopropyl nitrate and nitric acid were passed through the heated-inlet (bypassing the denuder) while the temperature was varied and NO2 was monitored"

This section describes how the experiment was conducted and should be moved to the experimental section, and not appear under "Results and discussion"

We restricted the experimental section to basic features of the instrument that apply to all laboratory and field experiments. For better readability of the paper we prefer to add short (section specific) experimental details in the results section.

pg 7 line 20. ~400 C to dissociate PAN is very high. How and where exactly is this temperature measured? What is the difference between the measured temperature and the temperature of the gas stream?

The temperature is measured in a ceramic block that accommodates the quartz tube. See also the reply to major comment (1) (b) and the new Figure S3a).

pg 7 line 22 "We conclude that PAN is stoichiometrically converted to NO2" Just because the curve has flattened does not imply stoichiometric conversion. Has this statement been verified, e.g., by comparison to a CL NOy instrument or PAN-CIMS?

Both CL $NO_y$ instruments and PAN-CIMS also require calibration. For PAN-CIMS, this has previously been done using TD-instruments operating in the plateau range. It makes little sense to reverse the logic and use the PAN-CIMS to calibrate the TD-CRD.

pg 7 line 24 it is also higher than Day et al. (2002), Paul et al. (2010), Di Carlo et al. (2013) and Sadanaga et al. (2016) and inconsistent with CIMS inlet performance (Slusher et al., 2004; Zheng et al., 2011; Mielke and Osthoff, 2012; Phillips et al., 2013).

There is no reason to expect identical thermograms when comparing different experimental set ups with different oven-types, tubes diameters pressures and flows. Figure S3a) shows that the offset between the oven temperature readout and the approximate thermocouple measured temperature in the center of the gas stream is largest at lower temperatures (84 °C at 230 °C). This would contribute to the explain the particularly large "apparent" difference in temperature needed to thermally dissociate PAN in the present study. In addition, we emphasise that our goal is not to measure PANs, ANs etc separately. This instrument was designed to measure $NO_Y$ and overlap of the "broad" thermograms is of no consequence.

pg 7 sections "3.1.1. PAN" and 3.1.2 "Isopropyl nitrate". Please comment on the possibility that the PAN and iPN sources contain impurities. PAN, for instance, will slowly decompose and form nitric acid and an alkyl nitrate.

We have remeasured the iPN thermogram with a new sample. The previously used iPN sample apparently contained a large $HNO_3$ impurity. See major comment (1) (a).

pg 7/8 "3.1.2 Isopropyl nitrate" and "3.1.3" I am skeptical about the accuracy of these thermograms. The overlap of iPN and HNO3 is inconsistent with literature.

See major comment (1) (a).

pg 8 line 3 "expected" Expected how?

We now write.

Based on the mixing ratio of iPN in the canister and the dilution flows, 10.7 ppbv represents (101 $\pm$ 11)% conversion.

pg 8 line 20. Please also compare with Di Carlo et al. (2013).

Reference to Di Carlo (2013) has been made.

pg 8 lines 22-25. If the temperatures are truly that inaccurate, please consider at least rough-calibrating the temperature scale.

As shown in Fig. S3a), the temperature measurement of the oven is clearly inaccurate. However, we do not have experimental means to accurately determine the temperature distribution at 850 °C set point over the whole length and width of the oven tube and therefore prefer to continue working with the nominal set point temperatures. We emphasize that almost all of the experiments that report thermograms rely on temperature measurement at the external surface of the inlet and not in the gas-phase.

pg 8 line 26. "Rather short" Why so short? In such a short inlet, the gas stream is likely heated very unevenly, leading to considerable broadening of the TD profiles. Was this intentional, or is this a design flaw? Note the broadening would not explain the overlap of iPN and HNO3 thermograms.

See new Sect. 3.4.

pg 8 line 29 if this calculation can be performed for 50% conversion, it can also be done for 10%, 20%, 30%, etc. to construct an expected TD profile, which in all likelihood will be inconsistent with Figure 2.

50 % conversion is the metric we also used to compare with literature thermograms. The overlap of iPN and $HNO_3$ has been resolved, the width of the dissociation steps due to the short residence times has been discussed above.

pg 8 line 32 "to ensure complete conversion of each trace gas to NO2". Replace complete with maximum conversion.

Corrected.

pg 9 line 4 "verify quantitative ... conversion to NO2". Quantitative conversion was not demonstrated in this work; In fact, Figure S2 suggests incomplete conversion (~85%; or 13/15 for HNO3).

See answer to major comment (1) (c). We write:

The $HNO_3$ thermogram (Fig. 2 and Fig. S2c)) has a plateau at temperatures above ≈ 800 °C. In the plateau region of Fig. 2, the $HNO_3$ mixing ratio measured is 13.0 ± 0.8 ppb, which (within combined uncertainties) is in agreement with the expected value (15.2 ± 1.98 ppb) calculated from the permeation rate and uncertainty in the dilution factor.

pg 9 line 6. Section "3.1.4 NH3" Womack et al. (2017) showed NH3 to be non-issue under most conditions, which is the same conclusion reached here. Does this insignificant interference really warrant two full pages? Consider condensing.

We would prefer to keep this section as it is. The interference is suppressed in the ambient air samples which both Womack et al. (2017) and we used. However, as discussed in the paper, there might be ambient conditions under which it becomes significant. In addition to the findings of Womack et al. (2017) and Wild et al. (2014) we showed a strong linear relationship between the $NH_3$ interference and $O_3$ added, quenched the signal with individual organic species, and confirmed the influence of RH on the quenching in both artificial and ambient atmospheres.

If NH3 can be converted to NOx, what happens when NH4NO3 or (NH4)2SO4 aerosol are sampled?

See reply to major comment (6).

pg 9 line 22 "addition of 30 ppbv isoprene to zero-air did not significantly reduce the NH3-to-NO2 conversion efficiency under dry conditions, but reduced it by a factor of two when the RH was increased to 50%" In the preceding section, it was shown that NH3 converts when there is a lot of O3 present; please clarify if isoprene was added in the absence or presence of O3 (with which it would react)?

$O_3$ was added in all experiments on the quenching of the $NH_3$ interference, as the interference is barely detectable without $O_3$, as seen in the pure $NH_3$ thermogram. A note about the $O_3$ concentration was added:

We found that addition of 30 ppbv isoprene to zero-air (containing 330 ppbv $O_3$) did not significantly reduce the $NH_3$-to-$NO_2$ conversion efficiency under dry conditions, but reduced it by a factor of two when the RH was increased to 50%.

pg 11 R22. How much CH3C(O)O2NO2 are expected in a heated inlet? Its dissociation reaction is missing.

The dissociation reaction has been added as R22a.

We also explored the potential for bias caused by the recombination of $CH_3C(O)O_2$ and $NO_2$  (reaction R22b), following the thermal decomposition of PAN (reaction R22a).

pg 11 equation (2). Is this equation valid for the likely turbulent flow conditions in the inlet?

The calculation of the diffusion coefficient is not coupled to flow conditions. Equation 3 applies to laminar flows. Under turbulent conditions transfer to the walls is likely to be more rapid.

pg 12 line 24/26 "Particle loss calculator (PCL)" Should this be PLC?

Corrected

pg 12 line 26 "which was developed for cylindrical piping and not the square honeycomb shape of the denuder and also does not take into account losses due to impact at the finite surface area which the gas/particle flow is exposed to at the entrance to the honeycomb" Based on this statement, wouldn't it be reasonable to conclude that this calculator should not be used?

The PLC gives a rough guide and indicates that significant particle loss is likely to occur for super-micron particles. We write:

The PLC does a better job in predicting a reduction in transmission for the largest particles which we measured and indicates a transmission of 74% at 1 μm and 45% at 2 μm. In certain environments, nitrate associated with coarse mode particles represents a potential (negative) bias for TD-CRDS measurements of $NO_y$.

pg 13 line 6. "close to 100%". Please provide a table with the precise values and some statistics.

Table S2 has been added.

The precise values from which the removal efficiencies in Fig. 6 were determined are listed in Table S2.

"humidified significant" Grammar.

Corrected.

"RH-dependent breakthrough". During zeroing, dry air is added and some of it travels through the denuder; does the denuder require some conditioning then after the switch back to ambient air sampling?

Zero air is never flowing back through the denuder to avoid exposing it (and fittings) to hot air. The valves are switched during zeroing to guarantee that excess zero air leaves towards ambient or through the non-denuder oven.

pg 13 line 9. ">95%. Please give precise values. Since NOx is usually the major component of NOy, the partial and variable (as a function of RH) transmission of NOx introduces a major bias when quantifying aerosol nitrate.

See Table S2.

pg 13 line 12. Why dry nitrogen (if the behaviour is different at high RH)? Is this really equivalent to 1 month of sampling ambient air? How was 2.30ïC´ t'1017 molecules derived at?

Dry nitrogen is the dilution gas of the iPN cylinder. The number of molecules deposited was calculated from the flow of iPN, the time of exposure and the cylinder mixing ratio.

pg 13 line 20 "2.55x1015" what are the units here?

It is a number of molecules: We write:

During this experiment, 2.55 x $10^{15}$ molecules of $NO_x$ desorbed from the denuder, indicating that the major fraction of iPN molecules remained stored on the denuder surface upon humidification.

pg 13 line 23 "After loading the denuder with 5 sccm from a 0.831 ppm NO2 gas bottle for 4.8 days" NO2 cylinders usually co-emit HONO, HNO3 and NO. Was this considered?

No, impurities in the $NO_2$ cylinder were not considered, as $NO_2$ is presumably still the dominant component. In a separate experiment we also observed that deposited $HNO_3$ is not re-released as $NO_x$ upon humidification. In our discussion we speculate that HONO might be an intermediate in the formation of the released NO.

pg 13 line 24 "(a total of 7.60 x 1017 molecules deposited)" How this value determined?

An explanation has been added.

[…], derived from the flow rate, the exposure time and the gas bottle mixing ratio)

pg 15 line 14 sections 4.1 and 4.2 are not convincing since there is no independent measure of what to expect for pNit.

These sections illustrate the first deployment of the instrument and are not intended to provide validation of e.g. the pNit measurement by inter-comparison. In contrast, the case study in Sect. 4.1 serves to demonstrate that humidity related interferences in the pNit channel are likely to be a source of bias under field conditions. This information is important to those striving to measure pNit with similar denuders. See also reply to major comment (2) (c).

pg 16 line 10 Does the AMS quantify supermicron particles at all? A citation is needed.

A citation has been provided:

(Drewnick et al., 2005).

pg 17 line 18 "> 90% transmission for ammonium nitrate". Is this statement justified when the transmission varies with size as shown in Figure 5?

This is why we give a size range in this sentence. Between ~40 and 400 nm the transmission is consistently above 90 %.

pg 21 line 45. The accepted paper should be cited.
pg 22 line 28. The accepted paper should be cited.
pg 23 line 13 The paper by Womack et al. (2017) has been accepted and should be cited and not its discussion paper.
Corrected for all three papers.

pg 24 Figure 1. Please add more detail such as dimensions; for example, indicate the diameter of the critical orifice and air pressure, and show l and d.
The two orifices have diameters of $\approx$ 0.05 mm.
This information has been added to the caption.

Figure 1 shows that ambient air is drawn in through two valves ("V")? Are these described anywhere? How much of a pressure drop do they give? What is the internal surface made of (Teflon?)?
Information on the valves has been added. See above.
An electronic, PTFE 3-way valve (*Neptune Research, Inc.*, type 648T032, diameter 4.4 cm, height 5.2 cm, orifice diameter 4 mm) under software control switches between the two heated inlets, one of which is equipped with a denuder. Memory effects through the employment of this valve were not observed. Bypassing the valve under normal sampling conditions led to an 0.6 torr pressure change.

In Figure 1, "1/2 in PFA" should be in metric units; indicate if this is outer or inner diameter and how long this section is.
Corrected.

In Figure 1, what does the green line represent? A chamber wall?
That was indeed a left-over from a previous diagram. The line has been removed.

The Figure is inconsistent with the text as it shows a TD oven at 850 C which is roughly the same length as the denuder; in the text, the section actually heated is described as rather short or 3 cm long, and the denuder length is stated at 10 cm.
Dimensions of the heated inlets parts are now included in the figure. A note has been added in the caption that the figure is not to scale.

pg 25 Figure 2. The PAN and iPN thermograms are inconsistent with literature. The caption should state the level of O3 present for the NH3 experiment.
Literature inconsistencies are discussed above. $O_3$ level was added:
[…] $NH_3$ (without added $O_3$).

pg 26 Figure 3 is not essential to this paper and probably should be in the supplemental.
The linear correlation between added $O_3$ and the extent of the $NH_3$ interference is an important finding and one of the main novelties in the $NH_3$ section, compared to the study of Womack et al. (2017). Therefore, we would preferably keep the figure in the main manuscript.

pg 28 Figure 5. Please specify the type of aerosol diameter (geometric vs. mobility). State value of key parameters (flow rate, denuder diameter). Correct inconsistency between PLC and PCL.

Figure and caption amended accordingly.

An aerosol flow of 3.3 slm was directed through the denuder (diameter 3 cm, see Sect. 2.4) and subsequently a DMA sampled 0.3 slm from the stream exiting the denuder.

pg 29 Figure 6. Please zoom in to ~60% to 100% for NO and to 95% to 100% for the other gases. Often, NO and NO2 are the largest components of NOy. How accurate is the measurement of NOz species if ~5% of NO2 and ~35% of NO break through the denuder? How variable are the numbers shown in Figure 6 (add error bars)?

The Figure has been re/drawn with error bars and an inset. The direct breakthrough of NO and $NO_2$ at zero-humidity is indeed just as problematic as the humidity induced re-release of NO in zero air. See above for discussion of the usability of pNit measurements (major comment (2) (c)). See new table S2 for precise values and variability metrics.

pg 30 Figures 7. What do the blue shades represent and why does the RH change (state in caption). As stated in the comments above, I am not convinced this experiment provides relevant information for an ambient air measurement (conc. are simply too high).

The caption has been extended. We have replied above to the comment on the high mixing ratios used.

The blue shaded area signifies the period in which the inlet oven was heated to 850 °C. Changes in RH are achieved by flowing parts of the zero air stream through deionized water.

$O_3$ addition was switched off during the blue shaded period.

pg 31 The presentation of Figure 8 is unclear.
How this experiment conducted?
Is synthetic air equal to zero air (and why is it humid then)?
In panel (a), why is there a "bump" at 9:50?
In panel (b), what does the derivative mean to the reader? What time?
Why are there lines for different RH?

The caption has been extended. The discontinuity ("bump") during the drying phase is caused by the presence of different adsorption sites and/or phase transitions on the denuder surface (as discussed in Sect. 3.3.2). The derivative was calculated in order to locate the position of this feature. Several RHs were plotted to demonstrate, that the discontinuity is slightly shifted to higher RH when starting the drying from a higher RH.

a) RH of humidified  zero air after passing through the denuder. The initial RH was determined by bypassing the denuder before and after the experiment. Zero air was humidified by flowing a fraction of the stream through deionized water stored in a glass vessel. The time at which the experiment was conducted is given on the x-axis. Until ca. 09:35 UTC air with constant humidity (RH ca. 68%) was send through the denuder. Behind the denuder, the measured humidity increased with a delay. Afterwards the denuder was exposed to dry zero air. b) Derivative of the measured RH during the drying period. The step during the drying phase occurs in a higher RH area, when starting the drying from a larger RH value.

pg 32 Figure 9. Properly cite Kim et al. in the bibliography, and do not provide the full reference in the caption.

Corrected.

pg 33 Figure 10a. Is the slope correct? There seem to be many points above the line. Is the slope affected by outliers? In any case, it's great that NOx data agree, but since the focus of this paper is mainly on measurement of NOy, NOz and particle nitrate, a more relevant plot would be TD-CRDS NOy vs. CLD NOy, TD-CRDS NOz vs. NOz measured by other techniques, as well TD-CRDS nitrate vs PILS or MARGA nitrate.

We have added a histogram of the $NO_x$ data points as Fig. S6. The histogram shows that 92 % of all data points are below 5 ppbv. These data points determine the slope of the overall correlation, explaining the visual misinterpretation caused by the relatively few data points between 10 and 25 ppbv. No other $NO_z$ or $NO_y$ measurements with which to compare were available during AQABA.

pg 33 Figure 10b I am not sure the OPC data add anything of value here since the nitrate fraction could be changing.

The OPC data set was added to exclude the possibility that the short term fluctuations in pNit were caused by coarse mode nitrates, undetected by the AMS, despite the non-nitrate-specific nature of the OPC signal.

Supplemental:

Figure S1. Please provide the mechanism used in the box model.

The reaction scheme has been added as table S1.

Figure S2, caption. "total uncertainty of the TD-CRDS measurements" - what is meant by this?

The caption has been refined.

Error bars represent the  measurement uncertainty of the TD-CRDS  (see Sect. 2.2).

Figure S2D is inconsistent with Figure 2 which shows the entire TD profile of iPN overlapping with that of HNO3 - which is not observed in Figure S2D under any condition.

The iPN thermogram has been re-measured. See above.

Figure S3. A strange and confusing way to present data.

The figure helps to explain the reason why addition of $O_3$ to convert NO to $NO_2$ removes the need for complex data correction.

The simulations consider oxidation of NO2 by O3 to form NO3 and subsequent formation of N2O5. Were NO3 and N2O5 sinks been considered? Please provide the full mechanism.

See new table S1.

**References**

Barker, J. R., Benson, S. W., Mendenhall, G. D., and Goldern, D. M.: Measurements of rate constants of importance in smog, Rep. PB-274530, Natl. Tech. Inf. Serv., Springfield, Va., 1977.

Baulch, D. L., Duxbury, J., Grant, S. J., and Montague, D. C.: Evaluated kinetic data for high temperature reactions. Volume 4 Homogeneous gas phase reactions of halogen- and cyanide- containing species, J. Phys. Chem. Ref. Data, 10, 1981.

Bridier, I., Caralp, F., Loirat, H., Lesclaux, R., Veyret, B., Becker, K. H., Reimer, A., and Zabel, F.: Kinetic and Theoretical-studies of the Reactions $CH_3C(O)O_2 + NO_2 + M <-> CH_3C(O)NO_2 + M$ between 248 K and 393 K and Between 30-torr and 760-torr, Journal of Physical Chemistry, 95, 3594-3600, 1991.

Day, D. A., Wooldridge, P. J., Dillon, M. B., Thornton, J. A., and Cohen, R. C.: A thermal dissociation laser-induced fluorescence instrument for in situ detection of $NO_2$, peroxy nitrates, alkyl nitrates, and $HNO_3$, J. Geophys. Res. -Atmos., 107, doi:10.1029/2001jd000779, 2002.

Drewnick, F., Hings, S. S., DeCarlo, P., Jayne, J. T., Gonin, M., Fuhrer, K., Weimer, S., Jimenez, J. L., Demerjian, K. L., Borrmann, S., and Worsnop, D. R.: A new time-of-flight aerosol mass spectrometer (TOF-AMS) - Instrument description and first field deployment, Aerosol Science and Technology, 39, 637-658, 10.1080/02786820500182040, 2005.

Glänzer, K., and Troe, J.: Thermal Decomposition of Nitrocompounds in Shock Waves. IV: Decomposition of Nitric Acid, Berichte der Bunsengesellschaft für physikalische Chemie, 78, 71-76, 10.1002/bbpc.19740780112, 1974.

IUPAC: Task Group on Atmospheric Chemical Kinetic Data Evaluation, (Ammann, M., Cox, R.A., Crowley, J.N., Herrmann, H., Jenkin, M.E., McNeill, V.F., Mellouki, A., Rossi, M. J., Troe, J. and Wallington, T. J.) http://iupac.pole-ether.fr/index.html., 2019.

Sobanski, N., Schuladen, J., Schuster, G., Lelieveld, J., and Crowley, J. N.: A five-channel cavity ring-down spectrometer for the detection of $NO_2$, $NO_3$, $N_2O_5$, total peroxy nitrates and total alkyl nitrates, Atmos. Meas. Tech., 9, 5103-5118, 10.5194/amt-9-5103-2016, 2016.

Thieser, J., Schuster, G., Phillips, G. J., Reiffs, A., Parchatka, U., Pöhler, D., Lelieveld, J., and Crowley, J. N.: A two-channel, thermal dissociation cavity-ringdown spectrometer for the detection of ambient $NO_2$, $RO_2NO_2$ and $RONO_2$, Atmos. Meas. Tech., 9, 553-576, 10.5194/amt-9-553-2016, 2016.

Tsang, W., and Herron, J. T.: Chemical kinetic data base for propellant combustion. I. Reactions involving NO, NO2, HNO, HNO2, HCN and N2O, JPCRD, 20, 609-663, 1991.

Wild, R. J., Edwards, P. M., Dube, W. P., Baumann, K., Edgerton, E. S., Quinn, P. K., Roberts, J. M., Rollins, A. W., Veres, P. R., Warneke, C., Williams, E. J., Yuan, B., and Brown, S. S.: A measurement of total reactive nitrogen, NOy, together with $NO_2$, NO, and $O_3$ via cavity ring-down spectroscopy, Environmental Science & Technology, 48, 9609-9615, doi:10.1021/es501896w, 2014.

Womack, C. C., Neuman, J. A., Veres, P. R., Eilerman, S. J., Brock, C. A., Decker, Z. C. J., Zarzana, K. J., Dube, W. P., Wild, R. J., Wooldridge, P. J., Cohen, R. C., and Brown, S. S.: Evaluation of the accuracy of thermal dissociation CRDS and LIF techniques for atmospheric measurement of reactive nitrogen species, Atmospheric Measurement Techniques, 10, 1911-1926, 10.5194/amt-10-1911-2017, 2017.

---

## Author Comment (AC3) · 17 Aug 2020

**Author's Response to Referee #3**

In this response, the referee comments (in black) are listed together with our replies (in blue) and the changes to the original manuscript (in red).

This manuscript does an overall good job of describing an instrument designed to measure NOx, NOy, and particulate nitrate by thermal dissociation – CRDS. The most useful aspect of the work is the demonstration of problems with the use of activated carbon denuders for removing gas-phase NOy compounds. This will be of great use to many other researchers who use these types of denuders and activated carbon in general!

I recommend it be published after addressing the minor comments below.

We thank the referee for the positive review of our paper and the helpful comments which we address in this response.

The detection limits are listed in the abstract (98 ppt for NOx with 1 min averaging) but strangely are not described elsewhere in the manuscript. Is this for a signal-to-noise ratio of 2? 3? How the LOD is defined and these numbers are determined should be in the main text somewhere. Given how sensitive CRDS can be to NO2, I am surprised that the LODs are as high as they are – I would have expected that with a minute of averaging the LOD would be quite a bit lower. Is this a result of the large correction (116 ppt) that must be made to account for the difference in Rayleigh scattering when sampling humid ambient air vs. dry zero air? In addition to that correction that must be made to account for the differences in humidity between sampling and zero measurements, doesn't the change in humidity also change the reflectivity of the mirrors (due to the change in the index of refraction of air), and thus the ring-down times?

We added a paragraph in Sect. 2.2 about the performance of the instrument and a critical comparison with other instruments (including LODs) in Sect. 3.4. The LOD we listed previously was from the AQABA campaign, where the ship's motions caused significant fluctuations in the ring-down times. We now list the performance obtainable on a stationary platform.

For NO2, the performance of the instrument was first described by Thieser et al. (2016), who reports a measurement uncertainty of 6 % + (20 pptv\*RH/100) which is dominated by uncertainty in the effective cross section of NO2 and the wavelength stability of the laser diode. The NOx detection limit of 40 pptv ( $2\sigma$ , 1 minute average) for the present instrument (laboratory conditions) was derived from an Allan variance analysis and is worse than that reported by Thieser et al. (2016) (6 pptv at 40 s) due to degradation of the mirror reflectivity. Corrections applied to take into account humidity and pressure changes are discussed in Sect. 2.1. The total uncertainty in NOy will depend on the uncertainty in the conversion to NOx of both gaseous and particulate nitrate and thus depends on the individual components of NOy in the air sampled. For purely gaseous NOy, the major problem is likely to be related to loss of sticky molecules at the inlet and we choose to quote a "worst case" uncertainty of 15%.

We have amended the LOD we quote to that obtained on a stationary platform (the one mentioned in the last version was derived from the AQABA dataset obtained on a ship):

In this context we note that the deployment on a ship resulted in a degradation in performance (LOD was  $\approx$  100 pptv) owing to the ship's motions, especially in heavy seas, which resulted in drifts in the instrument zero.

The zero air used for zeroes is "CAP 180, Fuhr GmbH"- please clarify what this meansis it compressed zero air from a cylinder, or is it from a zero air generator? Rather than deal with the effects of ambient sampling vs. dry zeroes, why not use humidity-matched air? (e.g., ambient air that has been scrubbed of NO2 via purafil or a catalyst?)

The zero-air generator has been described in more detail. Generally, the humidity correction is  $\leq$  100 pptv and has small associated uncertainties. Essentially we are scrubbing ambient air that has passed through a compressor.

 $k_0$  is typically determined every five minutes (for one minute) by overflowing the inlets with zero air from a commercial zero air generator (CAP 180, Fuhr GmbH) attached to a source of compressed ambient air.

pg 6, last line - define BET pg 13, "However, when the main dilution flow was humidified significant," This sentence appears to missing a word. Or perhaps the last word should actually be "significantly".

Both corrected.

[...] we calculate a BET (Brunauer-Emmett-Teller (Brunauer et al., 1938)) surface area [...]

[...] humidified significant [...]

**References**

Brunauer, S., Emmett, P. H., and Teller, E.: Adsorption of gases in multimolecular layers, J. Amer. Chem. Soc., 60, 309-319, 1938.

---

## Author Response (AR1)

**Measurement of NOx and NOy with a thermal dissociation cavity ringdown spectrometer (TD-CRDS): Instrument characterisation and first deployment.**

Nils Friedrich1, Ivan Tadic1, Jan Schuladen1, James Brooks2, Eoghan Darbyshire2, Frank Drewnick1, 5 Horst Fischer1, Jos Lelieveld1, John N. Crowley1

1Atmospheric Chemistry Department, Max Planck Institute for Chemistry, Mainz, 55128, Germany 2Centre of Atmospheric Science, University of Manchester, Manchester, UK

Correspondence to: John N. Crowley (john.crowley@mpic.de)

Abstract. We present a newly constructed, two channel Thermal Dissociation Cavity Ring-Down Spectrometer (TD-CRDS) for the detection measurement of NOx (NO + NO2), NOy (NOx + HNO3 + RO2NO2 + N2O5 etc.), NOz (NOy - NOx) and particulate nitrate (pNit). NOy-containing trace gases are detected as NO2 by CRDS at 405 nm following sampling through inlets at ambient temperature (NOx), or at 850 °C (NOy). In both cases, O3 was added to the air sample directly upstream of the cavities to convert NO (either ambient, or formed in the 850 °C oven) to NO2. An activated carbon denuder was used to remove gas-phase components of NOy when sampling pNit. Detection limits, defined as the 2σ precision for 1 minute

- 15 averaging, are 40 pptv for both NOx and NOy. The total measurement uncertainties (at 50% RH) in the NOx and NOy channels are 11% + 10 pptv and 16% + 14 pptv for NOz, respectively. Thermograms of various trace-gases of the NOz family confirm stoichiometric conversion to NO2 (and / or NO) at the oven temperature and rule out significant interferences from NH3 detection (< 2%) or radical recombination reactions under ambient conditions. While fulfilling the requirement of high particle transmission (> 80% between 30 and 400 nm) and essentially complete removal of reactive nitrogen under dry conditions (>
- 20 99%), the denuder suffered from NOx breakthrough and memory effects (i.e. release of stored NOy) under humid conditions, which may potentially bias measurements of particle nitrate. Summertime NOx measurements obtained from a ship sailing through the Red Sea, Indian Ocean and Arabian Gulf (NOx levels from < 20 pptv to 25 ppbv) were in excellent agreement with those taken by a chemiluminescence detector of NO and NO2. A dataset obtained locally under vastly different conditions (urban location in winter) revealed large diel variations in the NOz to
- 25 NOv ratio which could be attributed to the impact of local emissions by road-traffic.

**1** Introduction**

**1.1 Atmospheric NOx and NOy**

Total reactive nitrogen  $NO_y$  (=  $NO_x + NO_z$ ) consists of nitrogen oxide NO, nitrogen dioxide  $NO_2$  (NO +  $NO_2$  =  $NO_x$ ) and their reservoir species  $NO_z$  (NO3 +  $2N_2O_5$  +  $HNO_3$  + HONO +  $RONO_2$  +  $RO_2NO_2$  +  $XONO_2$  +  $XNO_2$  + pNit), where X is a halogen

- 5 atom. HCN and NH3 are generally not considered to be components of NOy (Logan, 1983). The formation of both peroxy nitrates (RO2NO2, PNs) and alkyl nitrates (RONO2, ANs) requires the presence of organic peroxy radicals (RO2), which are formed e.g. in the reaction of OH radicals with volatile organic compounds (VOCs) and oxygen (reaction R1). RO2 radicals react subsequently with NO2 or NO to form peroxy-nitrates (RO2NO2, PNs) or alkyl-nitrates (RONO2, ANs, reactions R2 and R3). Reaction R3 competes with the formation of an alkoxy radical (RO) and the oxidation
- 10 of NO to NO2 (reaction R4), which consumes the dominant fraction of RO2. The branching ratio between these two pathways depends on atmospheric conditions such as pressure and temperature and on the structure and length of the organic backbone (Lightfoot et al., 1992). HNO3 is produced mainly via the reaction of NO2 with OH (reaction R5).

$$OH + RH + O_2 \rightarrow RO_2 + H_2O$$
 (R1)

$$\begin{array}{ll} \text{RO}_2 + \text{NO}_2 + \text{M} & \rightarrow \text{RO}_2\text{NO}_2 + \text{M} & (\text{R2}) \\ \\ 15 & \text{RO}_2 + \text{NO} + \text{M} & \rightarrow \text{RONO}_2 + \text{M} & (\text{R3}) \\ & \text{RO}_2 + \text{NO} & \rightarrow \text{RO} + \text{NO}_2 & (\text{R4}) \\ & \text{OH} + \text{NO}_2 + \text{M} & \rightarrow \text{HNO}_3 + \text{M} & (\text{R5}) \end{array}$$

[revised manuscript text omitted]
 NH3 to NO2 conversion efficiency in zeroair containing O3. The addition of O3 results in a significant increase in NO2 with a linear dependence on the O3 mixing ratio (Fig. 3) with up to 11.4% conversion of NH3 to NO2 at 200 ppbv O3. This was not reduced measurably by the addition of water vapour to the air / O3 mixture. In further experiments we spiked air with the head-space of various organic liquids (acetone, methanol, ethanol, beta-pinene, limonene and isoprene). The gas-phase mixing ratios of the organic trace gases were unknown
- 25 but in each case the formation of NO2 was suppressed or completely stopped. A more quantitative investigation was carried out using a known concentration (1 ppmv gas bottle) of isoprene. We found that addition of 30 ppbv isoprene to zero-air (containing 330 ppbv O3) did not significantly reduce the NH3-to-NO2 conversion efficiency under dry conditions, but reduced it by a factor of two when the RH was increased to 50%.

A tentative chemical mechanism, based partially on Womack et al. (2017) to explain the formation of NO2 from NH3 and O3

at high temperatures and the processes that suppress it is given in reactions (R11 to R15). In this scheme, the oxidation of NH3 is initiated and propagated by O(3P), formed from the thermal dissociation of O3 (Peukert et al., 2013). This leads to formation of NO and HNO (R13a and R13b), both of which can be oxidised to NO2 (R14 and R15). Forward and reverse rate coefficients for reaction (R11) indicate that O3 is converted almost stoichiometrically to O(3P) in the  $\approx$  10 ms reaction time in the heated

inlet. The rate constants (at 1123 K) for the subsequent reactions involving O(3P) are:  $k_{12} = 4.4 \times 10^{-13} \text{ cm}^3 \text{ molec}^{-1} \text{ s}^{-1} k_{13a} = 8.3 \times 10^{-12} \text{ cm}^3 \text{molec}^{-1} \text{ s}^{-1}$  and  $k_{13b} = 7.5 \times 10^{-11} \text{ cm}^3 \text{ molec}^{-1} \text{ s}^{-1}$ ) (Cohen and Westberg, 1991). Reaction (R12) converts 0.3% of the initial NH3 molecules to NH2 within 10 ms (at 100 ppbv O3 and 1123 K).

5  $O_3 + M \rightarrow O(^3P) + O_2 + M$  (R11)

$$O(^{3}P) + NH_{3} \longrightarrow NH_{2} + OH$$
 (R12)

$$NH_2 + O(^{3}P) \longrightarrow NO + H_2$$

$$(R13a)$$

$$\rightarrow HNO + H$$

$$(R13b)$$

$$\rightarrow \Pi NO + \Pi$$
(R13)

(R14)

$$HNO + O(^{3}P) / OH / H \rightarrow NO + OH / H_{2}O / H_{2}$$

**$10 \quad \text{NO} + \text{O}(^3\text{P}) + \text{M} \rightarrow \text{NO}_2 + \text{M}$ (R15)**

The experimental results obtained in zero-air indicate that reactions involving  $O({}^{3}P)$  from  $O_{3}$  pyrolysis thermolysis can result in the conversion of NH3 to NO and NO2. 
[revised manuscript text omitted]
 NOx are reduced. The diel profiles of NOz/NOy are strongly influenced by fresh emissions of NOx. As the measurement location is

strongly influenced by traffic, there is a decrease in NOx (and increase in NOz/NOy) at nighttime. Nighttime increases in NOz ( $13^{th}-14^{th}$ ,  $15^{th}-16^{th}$ ,  $18^{th}-19^{th}$  and  $19^{th}-20^{th}$  of January 2020) may also be partially caused by formation of N2O5 as previously observed (Schuster et al., 2009) and which would have been favoured by the low nighttime temperatures (< 10 °C) in winter. These measurements serve to illustrate the applicability of our TD-CRDS over a wide range of NOx and NOy concentrations

5 under realistic field conditions and in the investigation of processes that transform NOx into its gas- and particle-phase reservoirs.

**5** Conclusions**

We report on the development, characterisation and first deployment of a TD-CRDS instrument for the measurement of NOx, NOy, NOz and pNit. Our laboratory experiments suggest that the different gas-phase NOz species investigated (PAN, iPN,

10 N2O5, HONO, ClNO2, HNO3) are converted with near stoichiometric efficiency to NOx at an oven temperature of 850 °C. NH4NO3 
[revised manuscript text omitted]